# Dimension-free error estimate for diffusion model and optimal scheduling

**Valentin de Bortoli**  *vdebortoli@google.com*
*Google DeepMind*

**Romuald Elie**  *relie@google.com*
*Google DeepMind*

**Anna Kazeykina**  *anna.kazeykina@universite-paris-saclay.fr*
*Université Paris Saclay*

**Zhenjie Ren**  *zhenjie.ren@univ-evry.fr*
*Université Evry - Paris Saclay*

**Jiacheng Zhang**  *jiachengzhang@cuhk.edu.hk*
*Chinese University of Hong Kong*

**Reviewed on OpenReview:** *https://openreview.net/forum?id=uArYtsvW8o*

## Abstract

Diffusion generative models have emerged as powerful tools for producing synthetic data from an empirically observed distribution. A common approach involves simulating the time-reversal of an Ornstein–Uhlenbeck (OU) process initialized at the true data distribution. Since the score function associated with the OU process is typically unknown, it is approximated using a trained neural network. This approximation, along with finite time simulation, time discretization and statistical approximation, introduce several sources of error whose impact on the generated samples must be carefully understood. Previous analyses have quantified the error between the generated and the true data distributions in terms of Wasserstein distance or Kullback–Leibler (KL) divergence. However, both metrics present limitations: KL divergence requires absolute continuity between distributions, while Wasserstein distance, though more general, leads to error bounds that scale poorly with dimension, rendering them impractical in high-dimensional settings. In this work, we derive an explicit, dimension-free bound on the discrepancy between the generated and the true data distributions. The bound is expressed in terms of a smooth test functional with bounded first and second derivatives. The key novelty lies in the use of this weaker, functional metric to obtain dimension-independent guarantees, at the cost of higher regularity on the test functions. As an application, we formulate and solve a variational problem to minimize the time-discretization error, leading to the derivation of an optimal time-scheduling strategy for the reverse-time diffusion. Interestingly, this scheduler has appeared previously in the literature in a different context; our analysis provides a new justification for its optimality, now grounded in minimizing the discretization bias in generative sampling.

## 1 Introduction

Generative models (GMs) have attracted substantial interest in machine learning over the past decade. Their core objective is to learn the underlying probability distribution of observed data, enabling the generation of new samples that resemble the training set. Notable classes of GMs include Variational Autoencoders (VAEs), Generative Adversarial Networks (GANs), and diffusion-based models (DMs), each employing dis-

tinct optimisation objectives to achieve faithful data synthesis and generalisation Goodfellow et al. (2014); Kingma & Welling (2013); Ho et al. (2020); Rezende et al. (2014).

Score-based generative diffusion models (SGMs) have recently gained particular attention, achieving state-of-the-art results in generative modelling Song & Ermon (2019); Song et al. (2021). Roughly speaking, the functioning of SGMs can be described as follows. Let $m_0$ denote the true data distribution that the model aims to learn. One first simulates an Ornstein–Uhlenbeck (OU) process starting from $m_0$; for sufficiently large times, its marginal distribution $m_t$ approaches the invariant measure of the OU process, denoted $m^*$, which is Gaussian. In the second step, a time-reversed process is simulated, initialised at $m^*$, to obtain an approximation of the original distribution $m_0$. However, the score $\nabla \ln m_t$ required for the time-reversal dynamics is unknown in practice and must be approximated, for example, by a neural network trained via score matching. Furthermore, a time-discretisation scheme is introduced to numerically simulate the reversed process.

This procedure inevitably introduces several sources of error: the *statistical error*, due to replacing the true data distribution with its empirical approximation from finite samples; the *finite-time simulation error*, stemming from approximating the true marginal distribution of the OU process by its invariant measure at large times; the *score-matching error*, arising from training the neural network to approximate the score function; and the *time-discretisation error*, incurred when simulating the reversed process numerically.

From a practical perspective, it is crucial to understand how these errors propagate through the diffusion model and impact the generated outputs. Most existing analyses of error estimation in diffusion models rely on relatively strong regularity assumptions on the score function or its estimator Chen et al. (2023d); De Bortoli (2023); De Bortoli et al. (2021); Chen et al. (2023a); Lee et al. (2022; 2023). More recent work Conforti et al. (2025) has proposed introducing an early stopping rule in the backward process to derive explicit and sharp bounds on the Kullback–Leibler (KL) divergence between the data distribution and the generative model output.

In the existing literature, the discrepancy between the generated distribution and the empirical data distribution is typically evaluated using the KL divergence or the Wasserstein distance, each of which has limitations. In particular, since the empirical approximation $m_0^N$ of the learned distribution $m_0$ is not absolutely continuous, the KL divergence is not applicable to quantify the statistical error. As for the Wasserstein distance, it yields an estimate of the form $W(m_0, m_0^N) = O(N^{-1/d})$, where $d$ is the data dimension Fournier & Guillin (2015). This curse of dimensionality on the estimate becomes very poor in high-dimensional settings common to real-world applications.

In this paper, we propose to overcome these limitations by introducing a weaker distance, defined through a class of smooth test functions with bounded derivatives. This approach yields statistical error estimates that are well-defined for empirical measures and free from the curse of dimensionality.

The main contribution of this paper is the derivation of a *dimension-free* explicit bound on the error between the true data distribution and the distribution generated by the SGM. This error is measured via a distance defined through a class of smooth test functions with bounded first and second derivatives. Achieving a dimension-free estimate comes at the cost of requiring higher regularity of the test functionals.

We begin by deriving the dimension-free error bound in the continuous-time setting. The resulting upper bound decomposes naturally into contributions from the statistical error, the finite-time simulation error, and the score-matching error incurred during the diffusion process. We then extend our analysis to incorporate a time discretisation scheme for simulating the time-reversal process. To improve the efficiency of the backward simulation, we introduce time scheduling into the time-reversal diffusion and derive the resulting error estimate as a function of both the discretisation step size and the scheduling function.

Importantly, minimising the derived error bound yields an *optimal scheduler* that enhances the accuracy of the generated approximation of the data distribution. This optimisation problem is equivalent to a variational problem for the scheduler, whose continuous-time solution can be obtained explicitly. Interestingly, the scheduler thus derived coincides with a scheduling scheme that has already been proposed in the SGM literature, where it has been empirically shown to improve generative performance Albergo et al. (2023b). Other scheduling strategies have also been studied previously Strasman et al. (2025). Our theoretical results

provide a rigorous justification for the optimality of the obtained scheduler, demonstrating that it minimises the error associated with the time discretisation of the inverse diffusion process.

## 2 Background

We first introduce diffusion model following the Stochastic Differential Equation (SDE) point of view of Song et al. (2021). We highlight that this is not the only possible presentation of diffusion models and more recent ones include Flow Matching Lipman et al. (2022) and Stochastic Interpolant Albergo et al. (2023a). However, focus on the SDE perspective as it is amenable to our theoretical investigation.

We aim at generating samples from a distribution $m_0$. Let $m_0$ denote the true underlying data distribution, and define the empirical distribution based on $N$ available data points as $m_0^N := \frac{1}{N} \sum_{i=1}^{N} \delta_{x_0^i}$, where the samples $x_0^i$ are assumed to be i.i.d. draws from $m_0$. Let $X$ be forward diffusion:

$$dX_t = b(X_t)dt + dW_t.$$

Denote by $m_t := \text{Law}(X_t)$ given $X_0 \sim m_0$, and by $m_t^N := \text{Law}(X_t)$ given $X_0 \sim m_0^N$.

In order to generate sample from the distribution $m_0$ we consider the *time-reversal* of $X$. Under mild assumptions, see Cattiaux et al. (2023) for instance, we have that the backward process $(\overleftarrow{X}_t)_{t \in [0,T]}$ is given by the following SDE

$$d\overleftarrow{X}_t = \left( -b(\overleftarrow{X}_t) + \frac{\alpha^2 + 1}{2} \nabla \log m_{T-t}^N(\overleftarrow{X}_t) \right) dt + \alpha d\overleftarrow{W}_t, \quad \text{for } \alpha \geq 0.$$

Denote by $\mu_t^N := \text{Law}(\overleftarrow{X}_t)$ given $\text{Law}(\overleftarrow{X}_0) = m_T^N$. In particular, note that $\mu_T^N = m_0^N$. Next, introduce the parametrized function $S$ to approximate the score function $\nabla \log m^N$:

$$\inf_{S \in \mathcal{S}} \int_\delta^T \int \left| \nabla \log m_t^N(x) - S_t(x) \right|^2 m_t^N(dx)dt. \tag{1}$$

In practice, $S$ is parameterized with an expressive neural network and the loss is minimized using equation 1 using Stochastic Gradient descent derived optimizers.

## 3 Main result

### 3.1 Continuous-time model

In this work, we are interested in the accuracy of the generation of equation 2. More precisely, we are interested in the error between the distribution of $\mu_{T-\delta}^*$, which correspond to the distribution of the generative process stopped at time $T - \delta$, and $m_0$, the underlying target data distribution. We begin by considering the case without time-discretization and extend our results to deal with the time-discretization in Section 3.2.

**Assumption 3.1.** Assume that the drift $b$ is Lipschitz, that the forward diffusion admits a unique invariant measure $m^*$, and that $H(m_t|m^*) \leq e^{-2\rho t}H(m_0|m^*)$ for some $\rho > 0$, where $H$ denotes the relative entropy (Kullback-Liebler divergence) and, for measures $\mu$ and $\nu$ such that $\mu \ll \nu$, is defined via

$$H(\mu \mid \nu) = \int \log \left( \frac{d\mu}{d\nu} \right) d\mu.$$

**Assumption 3.2.** Assume that there exists a score matching function $S^*$ such that

1. $x \mapsto S_t^*(x; m_0^N)$ is $L$-Lipschitz on $t \in [\delta, T]$;

2. The score matching error satisfies:

$$\mathbb{E}\left[\int_\delta^T \int \left|\nabla \log m_t^N(x) - S_t^*(x)\right|^2 m_t^N(dx)dt\right] \le \epsilon^2.$$

**Remark 3.3.** Although $m_t^N$ is singular at $t = 0$, the Hessian of its log-density is bounded on any interval $[\delta, T]$ with $\delta > 0$. In the Ornstein–Uhlenbeck case $b(x) = -x$, a direct computation gives

$$\left\|\nabla^2 \log m_t^N\right\|_{\mathrm{op}} \le \frac{1}{1 - e^{-2t}} + \frac{e^{-2t} \max_{i,j} \|x_0^i - x_0^j\|^2}{(1 - e^{-2t})^2}.$$

Consequently, for any $0 < \delta \le t \le T$, the right-hand side is finite, and the bound depends only on $\delta$, $T$ and the initial diameter $\max_{i,j} \|x_0^i - x_0^j\|$. Therefore, when using $S_t^*$ to approximate $\nabla \log m_t^N$, it is justified to require $S_t^*$ to be Lipschitz in $x$ uniformly for $t \in [\delta, T]$.

Now consider the diffusion driven by the score matching function:

$$d\overleftarrow{X}_t^* = \left(-b(\overleftarrow{X}_t^*) + \frac{\alpha^2 + 1}{2} S_{T-t}^*(\overleftarrow{X}_t^*)\right)dt + \alpha d\overleftarrow{W}_t, \quad \text{for } \alpha \ge 0. \tag{2}$$

Denote by $\mu_t^* := \mathrm{Law}(\overleftarrow{X}_t^*)$ given $\mathrm{Law}(\overleftarrow{X}_0^*) = m^*$. We are going to estimate the error between $\mu_T^*$ and $\mu_T^N = m_0^N$, and further obtain the error between $\mu_T^*$ and $m_0$.

Unlike the vast majority of related literature, which estimates the error between the target measure $m_0$ and the simulated measure $\mu_T^*$ under some well-known metrics or divergences, such as Wasserstein distance, total variation and Kullback–Leibler divergence, we shall consider the error through regular test functionals. As we shall see, the regularity of the test functionals helps to obtain a dimension-free upper bound on the error.

Let $\mathcal{P}_2$ be the set of probability measures on $\mathbb{R}^d$ with finite second moment, and $G : \mathcal{P}_2 \to \mathbb{R}$ be a test functional. We say that $G$ is linearly differentiable, if there exists a linear derivative $\frac{\delta G}{\delta m} : \mathcal{P}_2 \times \mathbb{R}^d \to \mathbb{R}$ such that for any $m, m' \in \mathcal{P}_2$

$$G(m') - G(m) = \int_0^1 \int_{\mathbb{R}^d} \frac{\delta G}{\delta m}\left(\lambda m' + (1 - \lambda)m, x\right)(m' - m)(dx)d\lambda.$$

Similarly, we may define the second order linear derivative $\frac{\delta^2 G}{\delta m^2}$. Further, we recall the definition of the intrinsic derivative:

$$D_m G(m, x) := \nabla \frac{\delta G}{\delta m}(m, x),$$

where $\nabla$ denotes as usual the gradient in $x$.

**Assumption 3.4.** Assume that $G$ admits bounded intrinsic derivatives $D_m G$ and $\nabla D_m G$, as well as bounded linear derivatives $\frac{\delta G}{\delta m}$ and $\frac{\delta^2 G}{\delta m^2}$.

**Remark 3.5.**

1. Here are some examples of linear derivatives. If $G(m) = \int f(x)m(dx)$, then, $\frac{\delta G}{\delta m} = f(x)$, For a cylindrical functional of the form

$$G(m) = F\left(\int f_1(x)m(dx), \int f_2(x)m(dx), .., \int f_k(x)m(dx)\right)$$

we have

$$\frac{\delta G}{\delta m} = \sum_{i=1}^k \partial_i F\left(\int f_1(x)m(dx), \int f_2(x)m(dx), .., \int f_k(x)m(dx)\right)f_i(x).$$

2. Let $r < d$ and let $A \in \mathbb{R}^{r \times d}$ satisfy $\|A\| \le 1$. For any $\phi \in C_b^2(\mathbb{R}^r)$, define

$$G(m) := \int_{\mathbb{R}^d} \phi(Ax)\, m(dx).$$

This functional admits bounded first and second linear and intrinsic derivatives with bounds not depending on the ambient dimension $d$. Since the metric only probes $m$ through test functions defined on a low-dimensional feature space, it can be insensitive to high-frequency differences in the ambient space. Concretely, two distributions can agree on low-complexity observables while differing on fine-scale structure (e.g. narrow spiky components, high-frequency texture, or perceptual artifacts).

**Theorem 3.6** (Error for continuous time-reversed diffusion). Let $S^*$ and $G$ satisfy Assumptions 3.1, 3.2 and 3.4. Then the error reads

$$\mathbb{E}\Big[\big|G(\mu^*_{T-\delta}) - G(m_0)\big|^2\Big] \leq C_1 \frac{1}{N} + C_2 e^{-2\rho T} + C_3 \epsilon^2 (\alpha^2 + 1)^2 + C_4 \delta^2,$$

where $C_1 = \tilde{C}_1 \|\frac{\delta G}{\delta m}\|_\infty \left(\|\frac{\delta^2 G}{\delta m^2}\|_\infty + \|\frac{\delta G}{\delta m}\|_\infty\right)$, $C_2 = \tilde{C}_2 \|\frac{\delta G}{\delta m}\|_\infty^2 H(m_0|m^*)$, $C_3 = \tilde{C}_3 T e^{CT} \|D_m G\|_\infty^2$, $C_4 = \tilde{C}_4 (\|D_m G\|_\infty^2 + \|\nabla \cdot D_m G\|_\infty^2)$, $\tilde{C}_1$, $\tilde{C}_2$, $\tilde{C}_3$, $\tilde{C}_4$ are independent constants, and $C$ depends linearly on the Lipschitz constants of $b$ and $S_t^*$.

**Remark 3.7.**

1. Note that $S^*$ and $\mu^*_{T-\delta}$ are random due to the randomness of $m_0^N$. The law $\mu^*_{T-\delta}$ is the law of the time-reversed diffusion process with the learned score function $S^*_{T-t}$ which is chosen to match the empirical score $m^N_{T-t}$ as in Assumption 3.2.

2. Note that a functional of the type $G(m) = \langle \phi, m \rangle$ for a given $\phi$ does not separate measures. However, one can take a supremum over all the $C^2$ bounded functions $\phi$ with bounded $\nabla \phi$, $\Delta \phi$ to obtain an error estimate on a functional that separates measures.

3. The proposed metrics is weaker than the Wasserstein metrics in the following sense. When measuring the distance between two measures in the Wasserstein-1 metrics, the difference between the measures is tested against Lipschitz functions with Lipschitz constant smaller than 1:

$$W_1(\mu, \nu) = \sup_{\|f\|_{Lip} \leq 1} \langle f, \mu - \nu \rangle.$$

   Note that $W_2$ is stronger, i.e. $W_1(\mu, \nu) \leq W_2(\mu, \nu)$. For metrics defined via G the testing is performed against a narrower class of regular test functions. To measure the TV distance between two functions one needs to consider even a larger set of test functions, namely, all bounded measurable functions: $\|\mu - \nu\|_{TV} = \frac{1}{2} \sup_{\|f\|_\infty \leq 1} \langle f, \mu - \nu \rangle$.

   The differences between the metrics considered above further leads to the following comparison between continuous measure $\mu$ with its empirical counterpart $\mu^N$ in terms of these metrics. Note, that due to the classical result of Fournier & Guillin (2015) we have $W_1(\mu, \mu^N) = O(1/N^{1/d})$. We also have that $\|\mu - \mu^N\|_{TV} = 1$ (does not decrease as N augments). At the same time, with the metrics considered in the present paper we get $G(\mu) - G(\mu^N) = O(1/\sqrt{N})$ (see Lemma 6.2).

   An example of a metric related to a functional $G$ satisfying the assumptions of the paper is the MMD metrics:

$$\text{MMD}_k^2(\mu, \nu) := \iint k(x, x')\, d\mu(x)\, d\mu(x') + \iint k(y, y')\, d\nu(y)\, d\nu(y') - 2 \iint k(x, y)\, d\mu(x)\, d\nu(y).$$

   If $k$ is, for example, a Gaussian kernal (often used in practice), then $G : \mu \mapsto \text{MMD}_k^2(\mu, \nu)$ admits bounded 1st and 2nd order linear and intrinsic derivatives.

4. Note that if the functional $G$ is chosen as in Remark 3.5 2, then the bounds on derivatives are dimension-independent; otherwise they typically scale linearly with $d$.

5. Theorem 3.6 gives a decomposition of error, notably, (i) statistical or finite-sample error given by $\frac{C_1}{N}$ term, (ii) finite time error $C_2 e^{-2\rho T}$, (iii) score matching error $C_3 \epsilon^2 (\alpha^2 + 1)$, and (iv) a small-time cutoff $C_4 \delta^2$.

   Given a desired error level err, one can take $\delta \leq \sqrt{\frac{\text{err}}{C_4}}$, $N \geq \frac{C_1}{\text{err}}$, and then choose $T$ of order $\frac{1}{2\rho} \log\left(\frac{\text{err}}{C_2}\right)$ and the score-matching error of order $\frac{1}{\alpha^2+1} \sqrt{\frac{\text{err}}{C_3}}$.

## 3.2 Discrete-time model and optimal scheduling

We further investigate the error when we discretize the time-reversal diffusion. In order to study the optimal discretization scheme, we introduce the scheduling $(g(t))_t$ of the forward diffusion:

$$dX_t = -\dot{g}(t)X_t dt + \sqrt{\dot{g}(t)}dW_t. \tag{3}$$

Compared to the study in Section 3.1, we restrict ourselves to the case $b(x) = -x$. Let $(m_t^N)_t$ be its marginal laws given $X_0 \sim m_0^N$. Note that the invariant measure $m^*$ is Gaussian $\mathcal{N}(0, \frac{1}{2}I_d)$ in this case.

Define $p_t^N := m_t^N/m^*$, so that

$$\nabla \ln p_t^N(x) := \nabla \ln m_t^N(x) + 2x.$$

The continuous time-reversal diffusion reads:

$$d\overleftarrow{X}_t = \dot{g}(T-t)\Big(-\overleftarrow{X}_t + \nabla \log p_{T-t}^N(\overleftarrow{X}_t)\Big)dt + \sqrt{\dot{g}(T-t)}d\overleftarrow{W}_t.$$

Denote as in Section 3.1 by $\mu_t^N := \text{Law}(\overleftarrow{X}_t)$ given $\text{Law}(\overleftarrow{X}_0) = m_T^N$. We are going to study the discretized time-reversal diffusion, where $\nabla \log p^N$ is approximated by the score function: for $t \in [t_k, t_{k+1})$

$$d\overleftarrow{X}_t^h = \dot{g}(T-t)\Big(-\overleftarrow{X}_t^h + S_{T-t_k}^*(\overleftarrow{X}_{t_k}^h)\Big)dt + \sqrt{\dot{g}(T-t)}d\overleftarrow{W}_t.$$

with $t_0 = 0$, $t_{k+1} - t_k = h$, $t_K = T - \delta$ and $\overleftarrow{X}_0^h \sim m^*$. Denote by $\mu_t^h := \text{Law}(\overleftarrow{X}_t^h)$.

**Assumption 3.8.** Assume that:

1. $x \mapsto S_t^*(x; m_0^N)$ is $L$-Lipschitz on $t \in [\delta, T]$;

2. The score matching error is well-controlled

$$\sum_{k=0}^{K-1} \int_{t_k}^{t_{k+1}} \dot{g}(T-t)dt \times \int \left|\nabla \log p_{T-t_k}^N(x) - S_{T-t_k}^*(x)\right|^2 m_{T-t_k}^N(dx) \leq \epsilon^2. \tag{4}$$

Note that we assume that for given $\epsilon$ and $g$ there exists a score-matching function $S^*$ such that the property (4) holds, i.e. $S^*$ depends on $\epsilon$ and $g$.

**Remark 3.9.** Compared to Section 3.1, we use the score function to approximate $\nabla \log p^N$ rather than $\nabla \log m^N$. This choice is motivated by two main reasons. First, when $b(x) = -x$, the invariant measure $m^*$ is explicitly known and Gaussian. Second, the proof of our discrete-time result relies on key estimates involving $\nabla \log p_{T-t}^N(\overleftarrow{X}_t)$. This dependence also explains why we restrict ourselves to the case $\alpha = 1$, in contrast to Section 3.1: for $\alpha \neq 1$, the definition of $\overleftarrow{X}$ changes, and the required estimate no longer holds.

Take a linear test functional

$$G(m) := \langle \varphi, m \rangle.$$

We consider the error between $G(\mu_{T-\delta}^h)$ and $G(m_0)$.

**Assumption 3.10.** Assume that $\varphi$ is bounded and of bounded derivatives $\nabla\varphi, \Delta\varphi$, and that $g$ is increasing and convex with $g(\delta) = \delta$.

**Theorem 3.11.** Let $G(m) = \langle \varphi, m \rangle$. Under Assumptions 3.8 and 3.10, we have

$$\mathbb{E}\Big[\big|G(\mu_{T-\delta}^h) - G(m_0)\big|^2\Big] \lesssim$$

$$(\|\nabla\varphi\|_\infty^2 + \|\Delta\varphi\|_\infty^2)\delta^2 + \|\varphi\|_\infty^2\Big(e^{-g(T)} + \frac{1}{N}\Big) + C_{g(T)}\|\nabla\varphi\|_\infty^2\epsilon^2 + \dot{g}(T)e^{2(\delta-g(T))} +$$

$$C_{g(T)}\|\nabla\varphi\|_\infty^2\frac{h}{\delta}\Big(\sum_{k=0}^{K-1}\int_{t_k}^{t_{k+1}}|\dot{g}(T-t)|^2 dt\, e^{2(\delta-g(T-t_{k+1}))}\Big).$$

**Remark 3.12.** Comparing the conclusions of Theorems 3.6 and 3.11, we see that the additional terms

$$\dot{g}(T)\, e^{2(\delta - g(T))} \;+\; Ch \sum_{k=0}^{K-1} \int_{t_k}^{t_{k+1}} |\dot{g}(T-t)|^2\, e^{-2g(T-t_{k+1})}\, dt$$

capture the discretization error. Since the first term is $O(e^{-2g(T)})$ and therefore negligible for sufficiently large $T$, controlling the discretization error reduces to choosing the scheduler $g$ so as to minimize

$$\inf_{g(\delta)=\delta,\, g(T)=T'} \sum_{k=0}^{K-1} \int_{t_k}^{t_{k+1}} |\dot{g}(T-t)|^2\, e^{-2g(T-t_{k+1})}\, dt.$$

Moreover, note that this discrete objective differs from its continuous-time counterpart by an $O(h)$ term. Indeed,

$$\sum_{k=0}^{K-1} \left| \int_{t_k}^{t_{k+1}} |\dot{g}(T-t)|^2 e^{-2g(T-t_{k+1})}\, dt - \int_{t_k}^{t_{k+1}} |\dot{g}(T-t)|^2 e^{-2g(T-t)}\, dt \right|$$

$$= \sum_{k=0}^{K-1} \int_{t_k}^{t_{k+1}} |\dot{g}(T-t)|^2 \left( e^{-2g(T-t_{k+1})} - e^{-2g(T-t)} \right) dt = O(h),$$

and a change of variables then yields

$$\sum_{k=0}^{K-1} \int_{t_k}^{t_{k+1}} |\dot{g}(T-t)|^2 e^{-2g(T-t)}\, dt = \int_0^T |\dot{g}(t)|^2 e^{-2g(t)}\, dt.$$

This motivates the following continuous-time optimal control problem:

$$\inf_{g(0)=0,\, g(T)=T'} \int_0^T |\dot{g}(t)|^2\, e^{-2g(t)}\, dt. \tag{5}$$

**Corollary 3.13.** The minimization problem in equation 5 is solved by the following optimal scheduling:

$$g^*(t) := -\ln\left( 1 - t\frac{1 - e^{-T'}}{T} \right).$$

## 4    Discussion on the optimal scheduler

We give in Corollary 3.13 the scheduler $g^*$ to minimizer the discretization error upper bound obtained in Theorem 3.11. Here we compute a Gaussian example, to illustrate that the scheduler is indeed often optimal.

Suppose the target distribution is $N(\mu, \sigma^2)$, that is $m_0 = N(\mu, \sigma^2)$. Then, the dynamic

$$dX_t = -\dot{g}(t) X_t + \sqrt{\dot{g}(t)}\, dW_t,$$

will have the distribution as

$$m_t = N\left( e^{-g(t)}\mu, \frac{1}{2} + e^{-2g(t)}\left( \sigma^2 - \frac{1}{2} \right) \right).$$

So we can compute $\nabla \log m_t(x)$ by

$$\nabla \log m_t(x) = -\frac{2(x - e^{-g(t)}\mu)}{1 + e^{-2g(t)}(2\sigma^2 - 1)}.$$

Therefore, assuming zero estimation error of the score function, we get the reverse discrete dynamic:

$$d\overleftarrow{X}_t^h = \dot{g}(T-t)\left( -\overleftarrow{X}_t^h - \frac{2(\overleftarrow{X}_{t_k}^h - e^{-g(T-t_k)}\mu)}{1 + e^{-2g(T-t_k)}(2\sigma^2 - 1)} + 2\overleftarrow{X}_{t_k}^h \right) dt + \sqrt{\dot{g}(T-t)}\, d\overleftarrow{W}_t$$

for $t \in [t_k, t_{k+1})$. Since inside each time interval, it is an OU process, we can explicitly write out the distribution of $\overleftarrow{X}^h_t$ given that $\mathrm{Law}(\overleftarrow{X}^h_0) = \mathrm{N}(0, \frac{1}{2} I_d)$.

In order to see the discretization error upper bound obtained in Theorem 3.11 c an be sharp, we take $\sigma^2 = \frac{1}{2}$. In this case, we have

$$\mathbb{E}\big[\overleftarrow{X}^h_{t_k}\big] = \mathbb{E}\big[\overleftarrow{X}^h_0\big]e^{-g(T)+g(T-t_k)} + 2\mu \sum_{\ell=1}^{k} e^{-g(T-t_\ell)+g(T-t_k)}\Big(e^{-g(T-t_{\ell-1})} - e^{-2g(T-t_{\ell-1})+g(T-t_\ell)}\Big)$$

$$= 2\mu e^{g(T-t_k)}\bigg(-\sum_{\ell=1}^{k} e^{-2g(T-t_{\ell-1})} + \sum_{\ell=1}^{k} e^{-g(T-t_{\ell-1})-g(T-t_\ell)}\bigg)$$

$$\approx 2\mu e^{g(T-t_k)} \sum_{\ell=1}^{k} e^{-2g(T-t_{\ell-1})}\dot{g}(T-t_{\ell-1})h. \quad (6)$$

Note that $\int_0^T e^{-2g(t)} 2\dot{g}(t)dt = 1 - e^{-2g(T)} \approx 1$. Therefore, the discretization error reads

$$\mu - \mathbb{E}\big[\overleftarrow{X}^h_T\big] \approx \mu \sum_{\ell=1}^{K} \int_{t_{\ell-1}}^{t_\ell} \big(e^{-2g(T-t)} - e^{-2g(T-t_{\ell-1})}\big) 2\dot{g}(T-t_{\ell-1})dt$$

$$\approx 4\mu h^2 \sum_{\ell=1}^{K} e^{-2g(T-t_{\ell-1})}|\dot{g}(T-t_{\ell-1})|^2,$$

where the right hand side is approximately the value we aim to minimize in Corollary 3.13. In this sense, we claim the scheduler in Corollary 3.13 can be optimal.

In the following we numerically evaluate the expectation $\mathbb{E}\big[\overleftarrow{X}^h_T\big]$ by equation 6 according to several common schedulers. Let $T = 1$, we shall try:

- The linear scheduler $g(t) = T't$;
- The optimal scheduler prescribed in Corollary 3.13 $g(t) = -\ln(1 - t(1 - e^{-T'}))$;
- The commonly used cosine scheduler $g(t) = -\ln(\cos(\frac{\pi}{2}at))$ where $a$ is such that $\cos(\frac{\pi}{2}a) = e^{-T'}$.

We compare the errors on the computed expectation values for different values of $\sigma$.

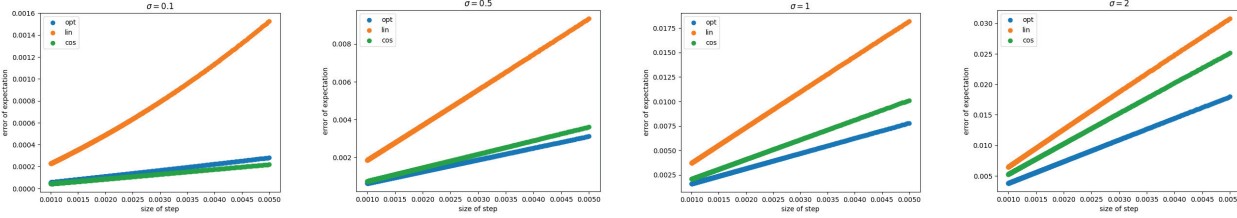

Figure 1: Error on expectation with different variances and schedulers

As shown in Figure 1, the prescribed optimal scheduler stays optimal or almost optimal in different cases.

Finally, we present a non-Gaussian experiment. We take $m_0$ to be a mixture of Gaussians in $\mathbb{R}^2$; in that case the score function can be computed explicitly. We compare $m_0$ and $\mu^h_{T-\delta}$ in terms of mean log-likelihood, Wasserstein W2, Wasserstein W1 and MMD metrics with a Gaussian kernel.

The results of the experiment are presented in Figures 2 and 3. We take $m_0 = 0.3N(0, I) + 0.7N((3, 3), 0.5I)$. We observe that the scheduler proposed in Corollary 3.13 significantly outperforms the linear scheduler, and generally exhibits superior performance compared to the cosine scheduler.

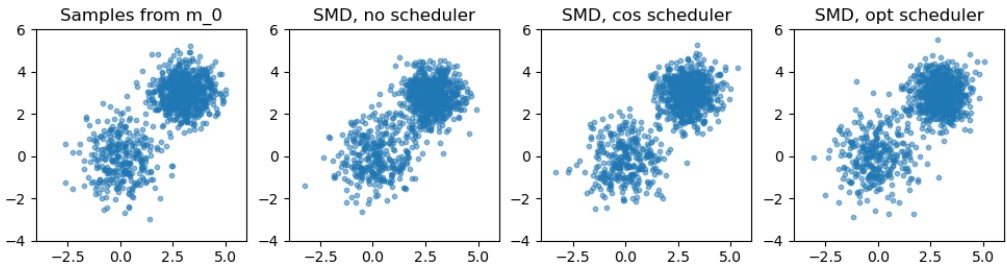

Figure 2: Target distribution $m_0$ and the result of score-matching diffusion (SMD) with linear, cosine and optimal scheduler.

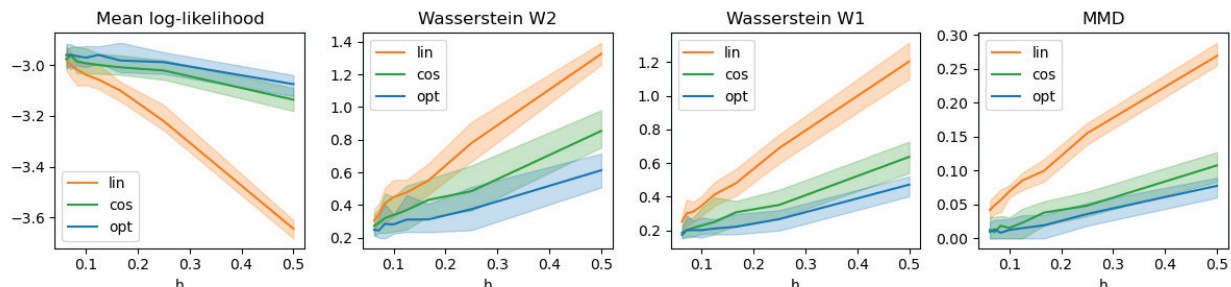

Figure 3: Comparison of $m_0$ and $\mu_{T-\delta}^h$ for a Gaussian mixture model in $\mathbb{R}^2$ (known score).

**Remark 4.1.**

1. For the dynamics of equation 3, the relative Fisher information with respect to the invariant Gaussian measure $m^*$ satisfies the sharp decay estimate

$$I(m_t \mid m^*) \approx e^{-2g(t)} I(m_0 \mid m^*),$$

   while the relative entropy $H(m_t \mid m^*)$ obeys the dissipation identity

$$dH(m_t \mid m^*) = -I(m_t \mid m^*) \dot{g}(t) \, dt \approx -I(m_0 \mid m^*) \, e^{-2g(t)} \dot{g}(t) \, dt.$$

   In our framework, the optimal scheduler $g^*$ is defined as a minimizer of equation 5. This choice enforces a uniform distribution in time of the effective entropy dissipation along the evolution on $[0, T]$.

   This criterion is closely related, at an intuitive level, to approaches that characterize optimal schedules via constant-speed geodesics in the Fisher–Rao metric Zhang & Syed (2025). While in Zhang & Syed (2025) the optimality is interpreted in terms of geometric interpolation in probability space, our perspective emphasizes temporal regularization of the dissipation rate. The two viewpoints coincide in special cases: for instance, when the initial law in equation 3 is Gaussian with the same variance as the invariant measure, $m_0 = N\left(\mu, \frac{1}{2}\right)$, the scheduler $g^*$ also yields a constant-speed Fisher–Rao geodesic. In general, however, the two notions of optimality do not coincide.

2. (Relation to variance-preserving and rectified-flow parameterizations) Consider the scheduled Ornstein–Uhlenbeck forward SDE

$$dX_t = -\dot{g}(t) \, X_t \, dt + \sqrt{\dot{g}(t)} \, dW_t,$$

   where $g : [0, T_{\mathrm{alg}}] \to [0, T']$ is increasing. Under the time change $s = g(t)$, this reduces to the standard OU process and yields the explicit coupling

$$X_t = e^{-g(t)} X_0 + \sqrt{1 - e^{-2g(t)}} \, Z.$$

Introducing the interpolation parameter $\tau_{\mathrm{VP}} := \sqrt{1 - e^{-2g(t)}}$, we recover the spherical (variance-preserving) interpolation $X_t = \sqrt{1 - \tau_{\mathrm{VP}}^2}\, X_0 + \tau_{\mathrm{VP}}\, Z$, which coincides with the VP coupling used in flow-matching.

Rectified flow instead employs the linear interpolation $\widetilde{X}_\tau = (1 - \tau)X_0 + \tau Z$. Defining $\tau_{\mathrm{RF}} := 1 - e^{-g(t)}$ linearizes the data coefficient $\alpha(t) = e^{-g(t)}$, but the corresponding noise coefficient becomes $\sqrt{\tau_{\mathrm{RF}}(2 - \tau_{\mathrm{RF}})}$, and thus does not match the linear RF path. Under the optimal scheduler $g^*(t)$ proposed in this work, $\alpha^*(t)$ is affine in algorithmic time, so that $\tau_{\mathrm{RF}}$ is linear in $t$. This is the sense in which the proposed scheduler aligns with rectified-flow parameterizations; the induced coupling remains the VP (spherical) interpolant rather than the linear RF path.

## 5    Related works

**State-of-the-art results**    There exists an extensive literature on the convergence of denoising diffusion models. While early works such as De Bortoli et al. (2021); Pidstrigach (2022); Block et al. (2020); De Bortoli (2023) and more recent works such as Choi & Fan (2025) proved convergence results with exponential dependencies in the problem parameters, those results have been refined in subsequent analyses.

Early works proved error bounds with polynomial dependencies with respect to the dimension or linear dependency with respect to the dimension but under strong assumptions on the regularity of the score function, see Li et al. (2023; 2024a); Lee et al. (2023); Chen et al. (2023d); Gupta et al. (2024). Those results were also extended to bound the probability flow ODE associated with diffusion models Chen et al. (2023c); Benton et al. (2023).

The first bounds on the Kullback-Leibler divergence *linear* with respect to the dimension were proven in Benton et al. (2024a); Conforti et al. (2025) and do not require any smoothness condition on the score. The dependency with respect to $\varepsilon$ was mitigated in Li & Yan (2024); Li et al. (2024b) to $O(d/\varepsilon)$. To the best of our knowledge the best results obtained so far are of complexity $O(\min(d, d^{2/3}L^{1/3}, d^{1/3}L)\varepsilon^{-2/3})$ in Jiao & Li (2024) leveraging a relaxed Lipschitz constant. It is also possible to improve those results by assuming that the data distribution is low dimensional Potaptchik et al. (2025); Liang et al. (2025). To the best of our knowledge, the result most closely related to our contribution is Li et al. (2025a) which identify dimension-free convergence rates for diffusion models in the case of Gaussian Mixture Models.

Those state-of-the-art results can even be improved with a dependency of $O(d^{1+2/K}\varepsilon^{-1/K})$ upon using higher order samplers Li et al. (2025b) and $O(d^{5/4}\varepsilon^{1/2})$ using accelerated samplers, see also Pedrotti et al. (2023). We also highlight that it is possible to improve those convergence rates even further, even deriving sub-linear time complexity results, by considering parallel sampling strategies as in Gupta et al. (2024); Bortoli et al. (2025). Those results have been partially extended to the case of stochastic interpolants Albergo et al. (2023b) in Liu et al. (2025) (with sub-optimal rates).

**Convergence in other modes.**    The convergence of the diffusion models regarding other metrics has been investigated in several papers. For instance in Mbacke & Rivasplata (2024); Cheng et al. (2024); Gao et al. (2023); Gao & Zhu (2025); Yu & Yu (2025) show the convergence of diffusion models with respect to the Wasserstein distance of order 2. Contrary to De Bortoli (2023) their results is not exponential in some parameters of the diffusion models. Other results on the convergence of diffusion models in Wasserstein distance including Lee et al. (2022) which improves significantly improves upon De Bortoli (2023). However it relies on strong assumptions regarding the Lipschitz constant of the denoiser. While Reeves & Pfister (2025) draws connection with information theory their analysis does not provide insights with respect to the discretization of the reverse process and its convergence.

**Other theoretical analyses.**    Finally, we highlight a few other areas of theoretical exploration of diffusion models. Li & Cai (2025) explored the convergence of discrete categorical diffusion models, while Biroli et al. (2024) characterized different modes of the backward diffusion leveraging tools from statistical physics. Chen et al. (2023b) provides sample complexity bounds for distribution estimation in low dimensional settings. Finally, Buchanan et al. (2025) identified memorization and generalization conditions for diffusion models. Chen et al. (2024); Gatmiry et al. (2024) focus on the *learning* problem in diffusion models and provide result

for general Gaussian mixtures. Combining learning guarantees with the aforementioned convergence results several works have derived near minimax optimality results Cai & Li (2025); Oko et al. (2023); Azangulov et al. (2025).

## 6  Proof of Theorem 3.6

Introduce

$$U(t,m) := G\big(\text{Law}(\overleftarrow{X}^*_{T-\delta}|\overleftarrow{X}^*_t \sim m)\big).$$

By definition of $U$ we have trivially that $U(T-\delta, \mu^*_{T-\delta}) = G\big(\text{Law}(\overleftarrow{X}^*_{T-\delta}|\overleftarrow{X}^*_{T-\delta} \sim \mu^*_{T-\delta})\big) = G(\mu^*_{T-\delta})$. Also $U(0, \mu^*_0) = G\big(\text{Law}(\overleftarrow{X}^*_{T-\delta}|\overleftarrow{X}^*_0 \sim m^*)\big) = G(\mu^*_{T-\delta})$ by (2), so that

$$U(T-\delta, \mu^*_{T-\delta}) = U(0, \mu^*_0).$$

On the other hand, the function $U$ satisfies the backward Kolmogorov equation:

$$\partial_t U + \int \left( D_m U\big(-b + \frac{\alpha^2+1}{2} S^*_{T-\cdot}\big) + \frac{\alpha^2}{2}\nabla \cdot D_m U \right) dm = 0 \tag{7}$$

(see Theorem 7.2 of Buckdahn et al. (2017)).

**Lemma 6.1** (Regularity bound for $U$)**.** Under Assumption 3.2 (1) and 3.4, we have the following regularity estimate on $U$:

$$\big|D_m U(t,m,x)\big| \leq C_T \|D_m G\|_\infty, \quad \left|\frac{\delta U}{\delta m}(t,m,x)\right| \leq C\left\|\frac{\delta G}{\delta m}\right\|, \quad \left|\frac{\delta^2 U}{\delta m^2}(t,m,x)\right| \leq C\left\|\frac{\delta^2 G}{\delta m^2}\right\|.$$

**Lemma 6.2.** Assume that $\varphi$ has bounded derivatives $\frac{\delta\varphi}{\delta m}$ and $\frac{\delta^2\varphi}{\delta m^2}$. Then, there exists a constant $C$, depending on the bounds on the first and second-order derivatives, such that,

$$\mathbb{E}^{(x_0^i)_i \sim m_0^{\otimes N}}\left[\big|\varphi(m_T) - \varphi(m_T^N)\big|^2\right] \leq \frac{C}{N}.$$

The proof of Lemma 6.1 and 6.2 is postponed to the Appendix.

*Proof of Theorem 3.6.* The weak error between $\mu^*_{T-\delta}$ and $\mu^N_{T-\delta} = m_\delta^N$ reads

$$\begin{aligned}
&G\big(\mu^*_{T-\delta}\big) - G\big(m_\delta^N\big) \\
=~& U\big(T-\delta, \mu^*_{T-\delta}\big) - U\big(T-\delta, \mu^N_{T-\delta}\big) \\
=~& U\big(0, m^*\big) - U\big(0, m_T^N\big) \\
&- \int_0^{T-\delta} \int \left( \partial_t U + D_m U\big(-b + \frac{\alpha^2+1}{2}\nabla\log m^N_{T-t}\big) + \frac{\alpha^2}{2}\nabla \cdot D_m U \right)\big(t, m^N_{T-t}, x\big) m^N_{T-t}(dx)dt \\
=~& U\big(0, m^*\big) - U\big(0, m_T^N\big) \\
&- \frac{\alpha^2+1}{2}\int_0^{T-\delta} \int \big(D_m U(\nabla\log m^N_{T-t} - S^*_{T-t})\big)\big(t, m^N_{T-t}, x\big) m^N_{T-t}(dx)dt,
\end{aligned}$$

where the second equality holds due to the Itô's formula for measure processes (see for instance Theorem 5.99 of Carmona & Delarue (2018)) and the third equality holds due to the backward Kolmogorov equation (7).

Recall Pinsker's inequality: $\|\mu - \nu\|^2_{TV} \leq \frac{1}{2}H(\mu \mid \nu)$, where $\|\mu - \nu\|_{TV}$ denotes the total variation distance between $\mu$ and $\nu$. This inequality implies that $|U\big(0, m_T\big) - U\big(0, m^*\big)|^2 \leq \frac{1}{2}\left\|\frac{\delta U}{\delta m}\right\|^2_\infty H(m_T|m^*)$.

Therefore, by Pinsker's inequality and Assumption 3.1, we have

$$\mathbb{E}\left[\left|G\left(\mu_{T-\delta}^{*}\right) - G\left(m_{\delta}^{N}\right)\right|^{2}\right]$$
$$\leq C\mathbb{E}\left[\left|U\left(0, m^{*}\right) - U\left(0, m_{T}^{N}\right)\right|^{2}\right] + C_{T}\epsilon^{2}(\alpha^{2} + 1)^{2}\|D_{m}U\|_{\infty}^{2}$$
$$\leq C\left\|\frac{\delta U}{\delta m}\right\|_{\infty}^{2}H(m_{T}|m^{*}) + C\mathbb{E}\left[\left|U\left(0, m_{T}\right) - U\left(0, m_{T}^{N}\right)\right|^{2}\right] + C_{T}\|D_{m}U\|_{\infty}^{2}\epsilon^{2}(\alpha^{2} + 1)^{2}$$
$$\leq C\left\|\frac{\delta U}{\delta m}\right\|_{\infty}^{2}e^{-2\rho T}H(m_{0}|m^{*}) + C\mathbb{E}\left[\left|U\left(0, m_{T}\right) - U\left(0, m_{T}^{N}\right)\right|^{2}\right] + C_{T}\|D_{m}U\|_{\infty}^{2}\epsilon^{2}(\alpha^{2} + 1)^{2},$$

where constant $C$ may vary from line to line. Moreover, due to the boundedness of the linear derivative of $G$ and $U(0, \cdot)$ (Lemma 6.1), by Itô's formula and Lemma 6.2 we obtain:

$$\mathbb{E}\left[\left|G\left(m_{0}\right) - G\left(m_{\delta}\right)\right|^{2}\right] = \mathbb{E}\left[\left|\int_{0}^{\delta}\int_{\mathbb{R}^{d}}\left(D_{m}G(m_{t}, x)b(x) + \frac{1}{2}\nabla \cdot D_{m}G(m_{t}, x)\right)m_{t}(dx)dt\right|^{2}\right] \leq C\delta^{2},$$

$$\mathbb{E}\left[\left|G\left(m_{\delta}\right) - G\left(m_{\delta}^{N}\right)\right|^{2}\right] \leq \frac{C}{N}, \qquad \mathbb{E}\left[\left|U\left(0, m_{T}\right) - U\left(0, m_{T}^{N}\right)\right|^{2}\right] \leq \frac{C}{N}.$$

Finally, the desired estimate follows. $\qquad\square$

# 7 Proof of Theorem 3.11

Let us define a family of functions $\{u^{(k)}(\cdot; x)\}_{\{k=0,1,2,..,K-1, x\in\mathbb{R}\}}$ and $\{v^{(k)}\}_{k=0,1,..,K}$ by

$$\text{For } t \in [t_{k}, t_{k+1}), \quad u^{(k)}(t, x; x_{k}) := \mathbb{E}\left[v^{(k+1)}(\overleftarrow{X}_{t_{k+1}}^{h})\Big|\overleftarrow{X}_{t}^{h} = x, \overleftarrow{X}_{t_{k}}^{h} = x_{k}\right], \tag{8}$$

with

$$v^{(K)}(x) := \varphi(x), \text{ and } v^{(k)}(x) := u^{(k)}(t_{k}, x; x), \text{ for } k = 0, 1, 2, .., K - 1.$$

**Lemma 7.1.** Given the definition of $\{v^{(k)}\}_{k=0,1,..,K}$ and $\{u^{(k)}\}_{k=0,1,2,..,K-1}$ above, fix $k \in \{0, 1, .., K - 1\}$, for all $x \in \mathbb{R}^{d}$, and $t \in [t_{k}, t_{k+1})$ we have

$$\|u^{(k)}(t, \cdot; x)\|_{\infty} \leq \|\varphi\|_{\infty}, \quad \|\nabla u^{(k)}(t, \cdot; x)\|_{\infty} \leq \|\nabla\varphi\|_{\infty}e^{(1+\|S^{*}\|_{\text{Lip},1,\infty})g(T)}.$$

Moreover, for all $k \in \{0, 1, .., K\}$

$$\|v^{(k)}\|_{\infty} \leq \|\varphi\|_{\infty}, \quad \|\nabla v^{(k)}\|_{\infty} \leq \|\nabla\varphi\|_{\infty}e^{(1+\|S^{*}\|_{\text{Lip},1,\infty})T}.$$

The proof is postponed to the Appendix.

Further, on $[t_{k}, t_{k+1})$ the function $u^{(k)}(\cdot; x_{k})$ solves the PDE:

$$\begin{cases} \partial_{t}u^{(k)}(t, x; x_{k}) + \frac{1}{2}\dot{g}(T - t)\Delta u^{(k)}(t, x; x_{k}) + \dot{g}(T - t)\left(x + S_{T-t_{k}}^{*}(x_{k})\right)\nabla u^{(k)}(t, x; x_{k}) = 0. \\ u^{(k)}(t_{k+1}, x; x_{k}) = v^{(k+1)}(x). \end{cases} \tag{9}$$

Since

$$v^{(k)}(x) = u^{(k)}(t_{k}, x; x) = \mathbb{E}\left[v^{(k+1)}(\overleftarrow{X}_{t_{k+1}}^{h})\Big|\overleftarrow{X}_{t_{k}}^{h} = x\right] = \mathbb{E}\left[\varphi(\overleftarrow{X}_{T-\delta}^{h})\Big|\overleftarrow{X}_{t_{k}}^{h} = x\right],$$

we get $v^{(0)}(x) = \mathbb{E}\left[\varphi(\overleftarrow{X}_{T-\delta}^{h})\big|\overleftarrow{X}_{0}^{h} = x\right]$, and $\varphi(x) = v^{(K)}(x)$. Therefore, we have

$$G(\mu_{T-\delta}^{h}) - G(m_{\delta}^{N}) = \mathbb{E}\left[\varphi(\overleftarrow{X}_{T-\delta}^{h})\right] - \langle\varphi, m_{\delta}^{N}\rangle = \langle v^{(0)}, m^{*}\rangle - \langle v^{(K)}, m_{\delta}^{N}\rangle.$$

Now we are ready to prove the main result on the discrete-time model.

*Proof of Theorem 3.11.* Note that

$$G(\mu_{T-\delta}^h) - G(m_\delta^N) = \langle v^{(0)}, m^* \rangle - \langle v^{(0)}, m_T^N \rangle + \langle v^{(0)}, m_T^N \rangle - \langle v^{(K)}, m_\delta^N \rangle.$$

By Pinsker's inequality and Lemma 6.2, the difference $\langle v^{(0)}, m^* \rangle - \langle v^{(0)}, m_T^N \rangle$ is controlled as follows:

$$\mathbb{E}^{(x_0^i)_i \sim m_0^{\otimes N}} \left[ \left| \langle v^{(0)}, m^* \rangle - \langle v^{(0)}, m_T^N \rangle \right|^2 \right] \tag{10}$$

$$\lesssim \left| \langle v^{(0)}, m^* \rangle - \langle v^{(0)}, m_T \rangle \right|^2 + \mathbb{E}^{(x_0^i)_i \sim m_0^{\otimes N}} \left[ \left| \langle v^{(0)}, m_T \rangle - \langle v^{(0)}, m_T^N \rangle \right|^2 \right]$$

$$\leq \|v^{(0)}\|_\infty^2 \mathrm{TV}(m^*, m_T)^2 + \mathbb{E}^{(x_0^i)_i \sim m_0^{\otimes N}} \left[ \left| \langle v^{(0)}, m_T \rangle - \langle v^{(0)}, m_T^N \rangle \right|^2 \right]$$

$$\lesssim \|v^{(0)}\|_\infty^2 \left( H(m_T | m^*) + \frac{1}{N} \right) \lesssim \|\varphi\|_\infty^2 \left( e^{-2g(T)} + \frac{1}{N} \right). \tag{11}$$

To estimate the difference $\langle v^{(K)}, m_\delta^N \rangle - \langle v^{(0)}, m_T^N \rangle$, it follows from Itô's formula and the PDEs equation 9 that

$$\langle v^{(K)}, m_\delta^N \rangle - \langle v^{(0)}, m_T^N \rangle = \mathbb{E}\left[ v^{(K)}(\overleftarrow{X}_{T-\delta}) - v^{(0)}(\overleftarrow{X}_0) \right] = \sum_{k=0}^{K-1} \mathbb{E}\left[ v^{(k+1)}(\overleftarrow{X}_{t_{k+1}}) - v^{(k)}(\overleftarrow{X}_{t_k}) \right]$$

$$= \sum_{k=0}^{K-1} \mathbb{E}\left[ u^{(k)}(t_{k+1}, \overleftarrow{X}_{t_{k+1}}; \overleftarrow{X}_{t_k}) - u^{(k)}(t_k, \overleftarrow{X}_{t_k}; \overleftarrow{X}_{t_k}) \right]$$

$$\overset{\text{Itô's}}{=} \sum_{k=0}^{K-1} \mathbb{E}\left[ \int_{t_k}^{t_{k+1}} \left( \partial_t u^{(k)}(t, \overleftarrow{X}_t; \overleftarrow{X}_{t_k}) + \dot{g}(T-t))\Delta u^{(k)}(t, \overleftarrow{X}_t; \overleftarrow{X}_{t_k}) \right. \right.$$

$$\left. \left. + \dot{g}(T-t)\left( -\overleftarrow{X}_t + \nabla \ln p_{T-t}^N(\overleftarrow{X}_t) \right) \nabla u^{(k)}(t, \overleftarrow{X}_t; \overleftarrow{X}_{t_k}) \right) dt \right]$$

$$\overset{\text{equation } 9}{=} \sum_{k=0}^{K-1} \mathbb{E}\left[ \int_{t_k}^{t_{k+1}} \dot{g}(T-t)\left( \nabla \ln p_{T-t}^N(\overleftarrow{X}_t) - S_{T-t_k}^*(\overleftarrow{X}_{t_k}) \right) \nabla u^{(k)}(t, \overleftarrow{X}_t; \overleftarrow{X}_{t_k}) dt \right]$$

where $\overleftarrow{X}_0 \sim m_T^N$. Therefore,

$$\left( \langle v^{(K)}, m_\delta^N \rangle - \langle v^{(0)}, m_T^N \rangle \right)^2$$

$$\leq \mathbb{E}\left[ \left\{ \sum_{k=0}^{K-1} \int_{t_k}^{t_{k+1}} \dot{g}(T-t)\left( \nabla \ln p_{T-t}^N(\overleftarrow{X}_t) - S_{T-t_k}^*(\overleftarrow{X}_{t_k}) \right) \nabla u^{(k)}(t, \overleftarrow{X}_t; \overleftarrow{X}_{t_k}) dt \right\}^2 \right]$$

$$\leq \mathbb{E}\left[ \left\{ \sum_{k=0}^{K-1} \int_{t_k}^{t_{k+1}} \dot{g}(T-t) \left\| \nabla \ln p_{T-t}^N(\overleftarrow{X}_t) - S_{T-t_k}^*(\overleftarrow{X}_{t_k}) \right\|^2 dt \right\} \right.$$

$$\left. \cdot \left\{ \sum_{k=0}^{K-1} \int_{t_k}^{t_{k+1}} \dot{g}(T-t) \left\| \nabla u^{(k)}(t, \overleftarrow{X}_t; \overleftarrow{X}_{t_k}) \right\|^2 dt \right\} \right].$$

By equation 22, we have

$$\sum_{k=0}^{K-1} \int_{t_k}^{t_{k+1}} \dot{g}(T-t) \left\| \nabla u^{(k)}(t, \overleftarrow{X}_t; \overleftarrow{X}_{t_k}) \right\|^2 dt \leq \|\nabla u^{(k)}\|_\infty^2 \int_{t_0}^{t_K} \dot{g}(T-t) dt \tag{12}$$

$$\leq g(T) \|\nabla \varphi\|_\infty^2 e^{(2+2\|S^*\|_{\mathrm{Lip}})g(T)}.$$

In order to deal with the first term, using the assumption equation 4 and the estimate (38) in Conforti et al. (2025), we get

$$\mathbb{E}\Big[\sum_{k=0}^{K-1}\int_{t_k}^{t_{k+1}}\dot{g}(T-t)\big\|\nabla\ln p_{T-t}^N(\overleftarrow{X}_t)-S_{T-t_k}^*(\overleftarrow{X}_{t_k})\big\|^2\mathrm{d}t\Big]$$

$$\leq 2\sum_{k=0}^{K-1}\int_{t_k}^{t_{k+1}}\dot{g}(T-t)\mathrm{d}t\cdot\mathbb{E}\Big[\big\|\nabla\ln p_{T-t_k}^N(\overleftarrow{X}_{t_k})-S_{T-t_k}^*(\overleftarrow{X}_{t_k})\big\|^2\Big]$$

$$+2\sum_{k=0}^{K-1}\mathbb{E}\Big[\int_{t_k}^{t_{k+1}}\dot{g}(T-t)\big|\nabla\ln p_{T-t}^N(\overleftarrow{X}_t)-\nabla\ln p_{T-t_k}^N(\overleftarrow{X}_{t_k})\big|^2\mathrm{d}t\Big]$$

$$\leq 2\epsilon^2+C\sum_{k=0}^{K-1}\mathbb{E}\left[\int_{t_k}^{t_{k+1}}\dot{g}(T-t)\big(\mathcal{I}(m_{T-t}^N|m^*)-\mathcal{I}(m_{T-t_k}^N|m^*)\big)\mathrm{d}t\right],$$

Moreover, as proved in Proposition 3 of Conforti et al. (2025), we have

$$\mathcal{I}(m_{T-t}^N|m^*)\leq e^{2(g(T-t)-g(T-s))}\mathcal{I}(m_{T-s}^N|m^*)\leq\mathcal{I}(m_{T-s}^N|m^*)\quad\text{for }t<s.$$

Therefore,

$$\sum_{k=0}^{K-1}\mathbb{E}\left[\int_{t_k}^{t_{k+1}}\dot{g}(T-t)\big(\mathcal{I}(m_{T-t}^N|m^*)-\mathcal{I}(m_{T-t_k}^N|m^*)\big)\mathrm{d}t\right]$$

$$\leq\mathbb{E}\left[\sum_{k=0}^{K-1}\big(g(T-t_k)-g(T-t_{k+1})\big)\big(\mathcal{I}(m_{T-t_{k+1}}^N|m^*)-\mathcal{I}(m_{T-t_k}^N|m^*)\big)\right]$$

$$\leq\mathbb{E}\left[\sum_{k=1}^{K-1}\big(g(T-t_{k+1})-2g(T-t_k)+g(T-t_{k-1})\big)\mathcal{I}(m_{T-t_k}^N|m^*)+\big(g(T-t_{K-1})-g(\delta)\big)\mathcal{I}(m_\delta^N|m^*)\right]$$

$$\leq\mathbb{E}\mathcal{I}(m_\delta^N|m^*)\left(\sum_{k=1}^{K-1}\big(g(T-t_{k+1})-2g(T-t_k)+g(T-t_{k-1})\big)e^{2(\delta-g(T-t_k))}+g(T-t_{K-1})-\delta\right)$$

$$=\mathbb{E}\mathcal{I}(m_\delta^N|m^*)\left(\sum_{k=0}^{K-1}\big(g(T-t_k)-g(T-t_{k+1})\big)\big(e^{2(\delta-g(T-t_{k+1}))}-e^{2(\delta-g(T-t_k))}\big)+\big(g(T)-g(T-h)\big)e^{2(\delta-g(T))}\right)$$

$$\leq\mathbb{E}\mathcal{I}(m_\delta^N|m^*)\left(\sum_{k=0}^{K-1}2\big(g(T-t_k)-g(T-t_{k+1})\big)^2e^{2(\delta-g(T-t_{k+1}))}+h\dot{g}(T)e^{2(\delta-g(T))}\right)$$

$$\leq h\mathbb{E}\mathcal{I}(m_\delta^N|m^*)\left(\sum_{k=0}^{K-1}2\int_{t_k}^{t_k+1}|\dot{g}(T-t)|^2\mathrm{d}t\,e^{2(\delta-g(T-t_{k+1}))}+\dot{g}(T)e^{2(\delta-g(T))}\right)$$

Together with equation 12, we finally have

$$\big(\langle v^{(K)},m_\delta^N\rangle-\langle v^{(0)},m_T^N\rangle\big)^2$$

$$\leq C_T\left(\epsilon^2+h\mathbb{E}\mathcal{I}(m_\delta^N|m^*)\big(\sum_{k=0}^{K-1}2\int_{t_k}^{t_k+1}|\dot{g}(T-t)|^2\mathrm{d}t\,e^{2(\delta-g(T-t_{k+1}))}+\dot{g}(T)e^{2(\delta-g(T))}\big)\right)\|\nabla\varphi\|_\infty^2.$$

and combining with equation 10, we have

$$\mathbb{E}^{(x_0^i)_i\sim m_0^{\otimes N}}\left[\big(G(\mu_{T-\delta}^h)-G(m_\delta^N)\big)^2\right]\lesssim\|\varphi\|_\infty^2\Big(e^{-g(T)}+\frac{1}{N}\Big)+C_T\|\nabla\varphi\|_\infty^2\epsilon^2$$

$$+C_T\|\nabla\varphi\|_\infty^2\mathbb{E}\mathcal{I}(m_\delta^N|m^*)h\Big(\sum_{k=0}^{K-1}\int_{t_k}^{t_k+1}|\dot{g}(T-t)|^2\mathrm{d}t\,e^{2(\delta-g(T-t_{k+1}))}+\dot{g}(T)e^{2(\delta-g(T))}\Big).$$

By convexity of $\mathcal{I}$ in the first argument we get that $\mathcal{I}\left(m_\delta^N \mid m^*\right) \le \frac{1}{N}\sum_{i=1}^N \mathcal{I}\left(m_\delta^{X_i} \mid m^*\right)$, where $m_\delta^{X_i}$ is the law of process (3) at time $\delta$ corresponding to $m_0 = \delta_{X_i}$. Then a direct calculation gives that $\mathbb{E}\mathcal{I}\left(m_\delta^{X_i} \mid m^*\right) \le C\left(\mathrm{Var}[X_i] + \frac{d}{\delta}\right)$.

Finally, the difference $G(m_\delta^N) - G(m_0^N)$ follows from Itô's formula, and we obtain the desired estimate. $\qquad\square$

*Proof of Corollary 3.13.* Denote $\mathcal{G}(g(t)) = \int\limits_0^T |\dot{g}(t)|^2 e^{-2g(t)}dt$. Consider a variation $\delta_t$ of $g(t)$ verifying $\delta_t(0) = 0, \delta_t(T) = 0$. Then we have

$$\mathcal{G}(g(t) + \delta_t) - \mathcal{G}(g(t)) = A_g(\delta_t) + o(\delta_t), \text{ where}$$

$$A_g(\delta_t) = \int_0^T 2\dot{g}(t)\dot{\delta}_t e^{-2g(t)}dt - \int_0^T 2|\dot{g}(t)|^2 \delta_t e^{-2g(t)}dt.$$

Integrating by parts and taking the boundary conditions into account, we obtain the following first order condition on the solution of the optimization problem $g_t^*$:

$$\ddot{g}_t^* - |\dot{g}_t^*|^2 = 0.$$

Integrating this ODE we obtain that $g_t^* = -\ln(-t + C_1) + C_2$. Finally, using the boundary conditions we obtain that $C_1 = \frac{T}{1 - e^{-T'}}$, $C_2 = \ln\left(\frac{T}{1 - e^{-T'}}\right)$, which yields the required formula. $\qquad\square$

**Remark 7.2.** Let $g_t^*$ be the optimal scheduler. Consider a small variation $\delta_t$ of $g_t^*$ verifying $\delta_t(0) = 0, \delta_t(T) = 0$. Then

$$\mathcal{G}(g_t^* + \delta_t) - \mathcal{G}(g_t^*) = e^{-C_2}\int_0^T (C_1 - t)^2 \dot{\delta}_t^2 dt + o(\delta_t^2).$$

This term gives a measure of sensitivity of the error with respect to deviations from $g^*$.

## 8 Conclusion

In this paper, we have proved the convergence of denoising diffusion models under a weaker distance than what is usually considered in the literature. More precisely, we have derived an upper-bound on the weak error of a discretization of the denoising process under some regularity assumptions on the test functions used in the computation of the weak errors and regularity assumptions on the noisy score of the empirical measure. Our main result is a *dimensionless* upper bound on the weak convergence error of diffusion models. To the best of our knowledge this bound is new. We show how our proof can be extended to cover the case of discrete diffusion models. Finally, we show that we recover the well-known Flow Matching (or Rectified Flow) schedule by optimizing the upper bound of our weak error analysis.

While our analysis is tight in the Euclidean case, in future work we would like to extend our weak error bounds to the case of general Denoising Markov Models combining our analysis with the infinitesimal generator framework of Benton et al. (2024b).

### Acknowledgements

A. K.'s research is supported by PEPR PDE-AI project. Z. R's research is supported by the Finance For Energy Market Research Center, the France 2030 grant (ANR-21-EXES-0003) and PEPR PDE-AI project.

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

## A  Analog in finite-state generative model setting

Recently, the finite-state generative model driven by the time-reversal Markov jump process has been earning an increasing interest. In this section, we briefly explain how our method can adapt to this context. Since it is not within the main scope of the paper, we will only focus on the continuous time model (without time discretization error) for an oversimplified one-token model.

Consider an example where the token can take values from $\mathcal{K} := \{0, 1\}$. As in the diffusion model, we initialize the forward process at the distribution $m_0 \in \mathbb{R}^2$ on $\mathcal{K}$. Eventually we shall sample according to $m_0$, using a time-reversal process. Let the forward process be a jump process with jump rate $\lambda$ and transition kernel $\tau(i, j) = \frac{1}{2}$ for $i, j \in \mathcal{K}$. In this setting, the marginal distribution $(m_t)_t$ evolves as the solution to the equation:

$$\frac{d}{dt}m_t(i) = \lambda\left(\frac{1}{2} - m_t(i)\right), \quad \text{for all } i \in \mathcal{K}.$$

Therefore $m_t(i) = e^{-\lambda t}m_0(i) + (1 - e^{-\lambda t})\frac{1}{2}$, and the marginal $m_t$ exponentially converges to the invariant measure, the uniform distribution on $\mathcal{K}$, denoted by $m^*$. As studied in Pham et al. (2025), by denoting the score function $s_t(i) := \frac{m_t(i) - m_t(1-i)}{m_t(i)}$, the time-reversal jump process is of the jump rate $\overleftarrow{\lambda}_t(i) := \frac{\lambda}{2}(2 - s_t(i))$ and of the transition kernel $\overleftarrow{\tau}_t(i, 1 - i) := \frac{\lambda}{2}\frac{1 - s_t(i)}{\overleftarrow{\lambda}_t(i)}$ and $\overleftarrow{\tau}_t(i, i) := \frac{\lambda}{2\overleftarrow{\lambda}_t(i)}$. Let the time-reversal jump process be initialized by the distribution $m_T$. One can verify that its marginal distribution is $\mu_t = m_{T-t}$. The dynamics of $(\mu_t)_t$ reads

$$\frac{d}{dt}\mu_t(i) = \frac{\lambda}{2}\Big(\big(1 - s_{T-t}(1-i)\big)\mu_t(1-i) - \big(1 - s_{T-t}(i)\big)\mu_t(i)\Big).$$

In practice, due to limited access to data, one may initialize the forward process by a distribution $m_0^N$ instead of $m_0$. We will denote by $(m_t^N)_t$ the marginal distribution of the associated forward jump process, and by $(\mu_t^N)_t$ that of the time-reversal process. On the other hand, one may use the data to estimate the score function $s_t$ by a parametrized function $s^*$ and try to control the error so that

$$\sum_{i=0,1}\int_0^T |s_t(i) - s_t^*(i)|^2 m_t^N(i)dt < \epsilon^2. \tag{13}$$

Denote by $(\mu_t^*)_t$ the marginal distribution of the time-reversal jump process driven by the function $s^*$ and starting from $m^*$ so that $\mu_0^* = m^*$ and

$$\frac{d}{dt}\mu_t^*(i) = \frac{\lambda}{2}\Big(\big(1 - s_{T-t}^*(1-i)\big)\mu_t^*(1-i) - \big(1 - s_{T-t}^*(i)\big)\mu_t^*(i)\Big). \tag{14}$$

Now, as in our result for the diffusion model, we aim at measuring the error of the generative model through a smooth test function $G : \mathbb{R}^2 \to \mathbb{R}$ by estimating $\big|G(m_0) - G(\mu_T^*)\big|$.

Similar to the diffusion case, we again rely on the function $U(t, m) := G(\mu_T^{*,t,m})$ where $\mu_T^{*,t,m}$ is the terminal value at time $T$ of the solution to the equation equation 14 given the initial condition $\mu_t := m$. We have $G(m_0^N) = U(T, m_0^N) = U(T, \mu_T^N)$, $G(\mu_T^*) = U(0, m^*)$, and more importantly that $U(t, x_0, x_1)$ solves the backward Komologrov equation

$$\partial_t U + \frac{\lambda}{2} \sum_{i=0,1} \partial_{x_i} U\Big(\big(1 - s_{T-t}^*(1-i)\big)x_{1-i} - \big(1 - s_{T-t}^*(i)\big)x_i\Big) = 0. \tag{15}$$

Now note that

$$G(m_0) - G(\mu_T^*) = G(m_0) - G(m_0^N) + U(T, \mu_T^N) - U(0, \mu_0^N) + U(0, m_T^N) - U(0, m^*).$$

While it is easy to obtain $\mathbb{E}|G(m_0) - G(m_0^N)| \lesssim \frac{1}{\sqrt{N}}$ and $\quad |U(0, m_T^N) - U(0, m^*)| \lesssim e^{-\lambda T}$, the estimate of the middle term relies on the PDE equation 15. Provided that $\partial_{x_i} U$ is bounded, we can verify that

$$|U(T, \mu_T^N) - U(0, \mu_0^N)| =$$

$$\frac{\lambda}{2}\Big|\int_0^T \sum_{i=0,1} \partial_{x_i} U(t, \mu_t^N)\Big(\big(s_{T-t}(1-i) - s_{T-t}^*(1-i)\big)\mu_t^N(1-i) - \big(s_{T-t}(i) - s_{T-t}^*(i)\big)\mu_t^N(i)\Big)dt\Big| \lesssim \epsilon T,$$

where the last inequality is due to assumption on the score matching error equation 13. Therefore, the conclusion reads

$$\mathbb{E}|G(m_0) - G(\mu_T^*)| \lesssim \frac{1}{\sqrt{N}} + e^{-\lambda T} + \epsilon T.$$

**Remark A.1.** The literature on error estimation for discrete-state diffusion models remains relatively sparse. We note in particular the following several works that contribute to the understanding of discrete space diffusion models. Su et al. (2025) derive bounds on TV distance between target and generated distibution in terms of statistical error in the framework of discrete flow matching. Huang et al. (2025) develop the complexity theory for (masked) discrete diffusion providing bounds on the number of discrete score evaluators necessary to achieve the desired TV accuracy. Wang et al. (2025) introduce training techniques for reasoning diffusion models that allow for estimating the KL divergence accuracy in an analytically tractable manner. Wan et al. (2025) develop a theoretical framework for deriving non-asymptotic KL divergence and TV error bounds for discrete flows with generator matching. Kumar et al. (2025) show that the visual autoregressive generation is mathematically a discrete diffusion model allowing to export diffusion-related tools to VAR.

The works that are closest to our approach are Pham et al. (2025); Li & Cai (2025), quantifying the error via the Kullback–Leibler (KL) divergence. Note, however, that when the state space $K$ is large and the token vocabulary is rich, then the empirical or model distribution $m_0$ is typically highly sparse. In that regime, the KL divergence is often ill-posed (infinite or numerically unstable due to zero coordinates) unless ad hoc smoothing is introduced. Measuring generalization error through a test regular function, as above, avoids this issue.

# B   Proofs of Lemmas

Here we collect the proofs for the technical lemmas.

## B.1   Proof of Lemma 6.1

*Proof of Lemma 6.1.* First, it follows directly from (Buckdahn et al., 2017, Lemma 5.1) that

$$D_m U(t, m, x) = \mathbb{E}\left[\partial_x \overleftarrow{X}_{T-\delta}^{*,t,x} \cdot D_m G\big(\mu_{T-\delta-t}^*(m), \overleftarrow{X}_{T-\delta}^{*,t,x}\big)\right],$$

where $\overleftarrow{X}^{*,t,x}$ the process defined in equation 2 starting from $x$ at time $t$. Then, the bound of the intrinsic derivative of $U$, $D_m U$, follows from our Lipschitz assumption on $b$ and $S^*$ together with the boundedness of $D_m G$.

In order to estimate the linear derivative, we perform the following lifting, enabling us to reuse the estimate for the intrinsic derivative. Define the operator $\mathcal{T} : \mathcal{C}(\mathbb{R}^d) \to \mathcal{C}(\mathbb{R}^{d+1})$ by

$$\varphi(x) \mapsto \mathcal{T}\varphi(x, y) := \varphi(x)y,$$

and the adjoint operator $\mathcal{T}^* : \mathcal{P}(\mathbb{R}^{d+1}) \to \mathcal{P}(\mathbb{R}^d)$ by

$$\rho \mapsto \mathcal{T}^*(\rho)(\mathrm{d}x) := \int_{\mathbb{R}} y\rho(\mathrm{d}x\mathrm{d}y).$$

Define $\widetilde{G} : \mathcal{P}(\mathbb{R}^{d+1}) \to \mathbb{R}$ by

$$\widetilde{G}(\rho) := G\big(\mathcal{T}^*(\rho)\big).$$

For $\rho_1, \rho_2 \in \mathcal{P}(\mathbb{R}^{d+1})$, it holds, by definition of linear derivative,

$$\widetilde{G}(\rho_1) - \widetilde{G}(\rho_2) = G\big(\mathcal{T}^*(\rho_1)\big) - G\big(\mathcal{T}^*(\rho_2)\big)$$
$$= \int_0^1 \Big\langle \frac{\delta G}{\delta m}\big(\lambda\mathcal{T}^*(\rho_1) + (1-\lambda)\mathcal{T}^*(\rho_2), \cdot\big), \mathcal{T}^*(\rho_1) - \mathcal{T}^*(\rho_2)\Big\rangle \mathrm{d}\lambda$$
$$= \int_0^1 \Big\langle \mathcal{T}\frac{\delta G}{\delta m}\big(\lambda\mathcal{T}^*(\rho_1) + (1-\lambda)\mathcal{T}^*(\rho_2), \cdot\big), \rho_1 - \rho_2\Big\rangle \mathrm{d}\lambda.$$

Hence, we have for all $\rho \in \mathcal{P}(\mathbb{R}^{d+1})$,

$$\frac{\delta\widetilde{G}}{\delta\rho}(\rho) = \mathcal{T}\frac{\delta G}{\delta m}(\mathcal{T}^*\rho),$$

that is for all $x \in \mathbb{R}^d, y \in \mathbb{R}$, we have

$$\frac{\delta\widetilde{G}}{\delta\rho}\big(\rho, (x, y)\big) = \frac{\delta G}{\delta m}\big(\mathcal{T}^*\rho, x\big)y. \tag{16}$$

Similarly, we have

$$\frac{\delta^2\widetilde{G}}{\delta\rho^2}\big(\rho, (x, y), (x', y')\big) = \frac{\delta G}{\delta m}\big(\mathcal{T}^*\rho, x, x'\big)yy'. \tag{17}$$

Consider the dynamic:

$$\begin{cases} \mathrm{d}\tilde{X}_t = \Big(-b(\tilde{X}_t) + \frac{\alpha^2+1}{2}S^*_{T-t}(\tilde{X}_t)\Big)dt + \alpha\overleftarrow{\mathrm{d}W}_t; \\ \mathrm{d}Y_t = 0. \end{cases}$$

and $\rho_t^*(\rho) = \mathrm{Law}(\tilde{X}_t, Y_t)$ given $\mathrm{Law}(\tilde{X}_0, Y_0) = \rho$. Define $\widetilde{U} : [0, T] \times \mathcal{P}(\mathbb{R}^{d+1}) \to \mathbb{R}$ by

$$\widetilde{U}(t, \rho) = \widetilde{G}\big(\rho^*_{T-\delta-t}(\rho)\big).$$

Let $\rho := m \otimes \delta_1$, where $\delta_1$ is the Dirac measure at 1. It is not hard to see that $\rho_t^*(\rho) = \mu_t^*(m) \otimes \delta_1$, and

$$\mathcal{T}^*(\rho) = m, \quad \mathcal{T}^*\big(\rho_t^*(\rho)\big) = \mu_t^*(m)$$

then

$$\widetilde{U}(t, \rho) = \widetilde{G}\big(\rho^*_{T-\delta-t}(\rho)\big) = G\big(\mu^*_{T-\delta-t}(m)\big) = U(t, m).$$

Then, by equation 16 and equation 17, we have

$$\frac{\delta\widetilde{U}}{\delta\rho}\big(t, \rho, (x, y)\big) = \frac{\delta}{\delta\rho}\widetilde{G}\big(\rho^*_{T-\delta-t}(\rho), (x, y)\big) = \frac{\delta}{\delta m}G\big(\mu^*_{T-\delta-t}(m), x\big)y = \frac{\delta U}{\delta m}(t, m, x)y,$$

and similarly

$$\frac{\delta^2\widetilde{U}}{\delta\rho^2}\big(t, \rho, (x, y), (x', y')\big) = \frac{\delta^2 U}{\delta m^2}(t, m, x, x')yy'.$$

Therefore, we establish the link between the linear derivatives $\frac{\delta U}{\delta m}, \frac{\delta^2 U}{\delta m^2}$ with the intrinsic derivatives $\partial_y \frac{\delta \widetilde{U}}{\delta \rho}, \partial_{yy'}^2 \frac{\delta^2 \widetilde{U}}{\delta \rho^2}$:

$$\frac{\delta U}{\delta m}(t, m, x) = \partial_y \frac{\delta \widetilde{U}}{\delta \rho}(t, m, (x, y)), \quad \frac{\delta^2 U}{\delta m^2}(t, m, x, x') = \partial_{yy'}^2 \frac{\delta^2 \widetilde{U}}{\delta \rho^2}(t, \rho, (x, y), (x', y')).$$

Finally, we again apply (Buckdahn et al., 2017, Lemma 5.1, Lemma 5.2) to obtain the desired bounds. □

### B.2 Proof of Lemma 6.2

Note that

$$\langle \phi, m_0^N \rangle = \frac{1}{N} \sum_{i=1}^{N} \phi(X^i), \quad \langle \phi, m_T^N \rangle = \frac{1}{N} \sum_{i=1}^{N} \mathbb{E}\big[\phi(X_T)|X_0 = X^i\big],$$

where $X^1, .., X^N$ are i.i.d copies following the law $m_0$.

*Proof of Lemma 6.2.* This essentially repeats the proof of Lemma 5.10 in Delarue et al. (2019). Since $\varphi$ is differentiable, we have the following expansion:

$$\varphi(m_T^N) - \varphi(m_T) = \int_0^1 \Big\langle \frac{\delta \varphi}{\delta m}\big(\lambda m_T^N + (1 - \lambda)m_T, \cdot\big), m_T^N - m_T \Big\rangle d\lambda$$

$$= \underbrace{\int_0^1 \Big\langle \frac{\delta \varphi}{\delta m}\big(\lambda m_T^N + (1 - \lambda)m_T, \cdot\big) - \frac{\delta \varphi}{\delta m}(m_T, \cdot), m_T^N - m_T \Big\rangle d\lambda}_{S_1} + \underbrace{\Big\langle \frac{\delta \varphi}{\delta m}(m_T, \cdot), m_T^N - m_T \Big\rangle}_{S_2}.$$

**Step 1.** We first deal with the term $S_2$ This may be rewritten as

$$S_2 = \frac{1}{N} \sum_{i=1}^{N} \left( \mathbb{E}\left[ \frac{\delta \varphi}{\delta m}(m_T, X_T) \Big| X_0 = X^i \right] - \widetilde{\mathbb{E}}\left[ \frac{\delta \varphi}{\delta m}(m_T, \widetilde{X}_T) \right] \right),$$

where under $\widetilde{\mathbb{E}}$, $d\widetilde{X}_t = b(\widetilde{X}_t)dt + d\widetilde{W}_t$ and $\text{Law}(\widetilde{X}_0) = m_0$. Therefore, we have

$$\mathbb{E}[S_2^2] = \text{Var}\left( \frac{1}{N} \sum_{i=1}^{N} \mathbb{E}\left[ \frac{\delta \varphi}{\delta m}(m_T, X_T) \Big| X_0 = X^i \right] \right) = \frac{1}{N^2} \sum_{i=1}^{N} \text{Var}\left( \mathbb{E}\left[ \frac{\delta \varphi}{\delta m}(m_T, X_T) \Big| X_0 = X^i \right] \right)$$

$$= \frac{1}{N} \left( \mathbb{E}\left[ \mathbb{E}\left[ \frac{\delta \varphi}{\delta m}(m_T, X_T) \Big| X_0 = X^i \right]^2 \right] - \widetilde{\mathbb{E}}\left[ \frac{\delta \varphi}{\delta m}(m_T, X_T) \right]^2 \right) \le \frac{\left\| \frac{\delta \varphi}{\delta m} \right\|_\infty^2}{N}.$$

**Step 2.** We turn to the term $S_1$, which we first rewrite in the form

$$S_1 = \frac{1}{N} \sum_{i=1}^{N} \int_0^1 \varphi_\lambda^i d\lambda,$$

where we define, for $i = 1, \cdots, n$ and $\lambda \in [0, 1]$,

$$\varphi_\lambda^i := \mathbb{E}\left[ \frac{\delta \varphi}{\delta m}\big(\lambda m_T^N + (1 - \lambda)m_T, X_T\big) - \frac{\delta \varphi}{\delta m}(m_T, X_T) \Big| X_0 = X^i \right]$$

$$- \widetilde{\mathbb{E}}\left[ \frac{\delta \varphi}{\delta m}\big(\lambda m_T^N + (1 - \lambda)m_T, \widetilde{X}_T\big) - \frac{\delta \varphi}{\delta m}(m_T, \widetilde{X}_T) \right],$$

where $\widetilde{\mathbb{E}}$ is taken independently with $m_T^N$ (viewing $m_T^N$ as non-random) and $\widetilde{X}_T$ is given by the dynamic $d\widetilde{X}_t = b(\widetilde{X}_t)dt + d\widetilde{W}_t$ with $\text{Law}(\widetilde{X}_0) = m_0$. Now, we have

$$S_1^2 \le \frac{1}{N^2} \int_0^1 \left( \sum_{i=1}^{N} \varphi_\lambda^i \right)^2 d\lambda = \frac{1}{N^2} \int_0^1 \left( \sum_{i=1}^{N} (\varphi_\lambda^i)^2 + 2 \sum_{i<j} \varphi_\lambda^i \varphi_\lambda^j \right) d\lambda.$$

Note that $|\varphi_\lambda^i| \leq 4\|\frac{\delta\varphi}{\delta m}\|_\infty$, we have

$$\frac{1}{N^2}\int_0^1 \sum_{i=1}^N (\varphi_\lambda^i)^2 \mathrm{d}\lambda \leq \frac{16\|\frac{\delta\varphi}{\delta m}\|_\infty^2}{N}.$$

It remains to bound $\varphi_\lambda^i \varphi_\lambda^j$ for $i \neq j$. Define $m_T^{N,-i,-j}$ by

$$\langle \phi, m_T^{N,-i,-j}\rangle := \frac{1}{N-2}\sum_{k\neq i,j} \mathbb{E}\big[\phi(X_T)|X_0 = X^k\big],$$

and $\varphi_\lambda^{i,-j,-k}$ by

$$\varphi_\lambda^{i,-j,-k} := \mathbb{E}\left[\frac{\delta\varphi}{\delta m}\big(\lambda m_T^{N,-j,-k} + (1-\lambda)m_T, X_T\big) - \frac{\delta\varphi}{\delta m}(m_T, X_T)\Big| X_0 = X^i\right]$$
$$-\widetilde{\mathbb{E}}\left[\frac{\delta\varphi}{\delta m}\big(\lambda m_T^{N,-j,-k} + (1-\lambda)m_T, \widetilde{X}_T\big) - \frac{\delta\varphi}{\delta m}(m_T, \widetilde{X}_T)\right].$$

Then, it is not hard to see that $\varphi_\lambda^{i,-i,-j}$ is conditionally independent of $\varphi_\lambda^{j,-i,-j}$ given $\{X_k\}_{k\neq i,j}$ and therefore

$$\mathbb{E}\big[\varphi_\lambda^{i,-i,-j}\varphi_\lambda^{j,-i,-j}\big] = 0. \tag{18}$$

Observe that $\varphi_\lambda^i - \varphi_\lambda^{i,-j,-k}$ can be calculated by

$$\mathbb{E}\left[\frac{\delta\varphi}{\delta m}\big(\lambda m_T^N + (1-\lambda)m_T, X_T\big) - \frac{\delta\varphi}{\delta m}\big(\lambda m_T^{N,-j,-k} + (1-\lambda)m_T, X_T\big)\Big| X_0 = X^i\right]$$
$$-\widetilde{\mathbb{E}}\left[\frac{\delta\varphi}{\delta m}\big(\lambda m_T^N + (1-\lambda)m_T, X_T\big) - \frac{\delta\varphi}{\delta m}\big(\lambda m_T^{N,-j,-k} + (1-\lambda)m_T, X_T\big)\right], \tag{19}$$

and

$$\frac{\delta\varphi}{\delta m}\big(\lambda m_T^N + (1-\lambda)m_T, X_T\big) - \frac{\delta\varphi}{\delta m}\big(\lambda m_T^{N,-j,-k} + (1-\lambda)m_T, X_T\big)$$
$$= \int_0^1 \Big\langle \frac{\delta^2\varphi}{\delta m^2}\big(\lambda\gamma m_T^N + \lambda(1-\gamma)m_T^{N,-i,-j} + (1-\lambda)m_T, X_T, \cdot\big), \lambda\big(m_T^N - m_T^{N,-i,-j}\big)\Big\rangle \mathrm{d}\gamma.$$

Recall that

$$\langle\phi, m_T^N - m_T^{N,-i,-j}\rangle := \frac{1}{N}\Big(\mathbb{E}\big[\phi(X_T)|X_0 = X^i\big] + \mathbb{E}\big[\phi(X_T)|X_0 = X^j\big]\Big)$$
$$-\frac{2}{N(N-2)}\sum_{k\neq i,j}\mathbb{E}\big[\phi(X_T)|X_0 = X^k\big],$$

we have

$$\left|\frac{\delta\varphi}{\delta m}\big(\lambda m_T^N + (1-\lambda)m_T, X_T\big) - \frac{\delta\varphi}{\delta m}\big(\lambda m_T^{N,-j,-k} + (1-\lambda)m_T, X_T\big)\right| \leq \frac{2\lambda\|\frac{\delta^2\varphi}{\delta m^2}\|_\infty}{N} + \frac{2\lambda\|\frac{\delta^2\varphi}{\delta m^2}\|_\infty}{N-2}$$
$$\leq \frac{8\|\frac{\delta^2\varphi}{\delta m^2}\|_\infty}{N},$$

where the last inequality holds because without loss of generality, we could assume $N \geq 3$ and $3(N-2) \geq N$. Plugging back into equation 19, we get

$$\big|\varphi_\lambda^i - \varphi_\lambda^{i,-j,-k}\big| \leq \frac{16\|\frac{\delta^2\varphi}{\delta m^2}\|_\infty}{N}$$

and together with equation 18, we have

$$
\left| \mathbb{E}\big[\varphi_\lambda^i \varphi_\lambda^j\big] \right| = \left| \mathbb{E}\big[|\varphi_\lambda^{i,-i,-j} \varphi_\lambda^{j,-i,-j}\big] + \mathbb{E}\Big[\varphi_\lambda^i (\varphi_\lambda^j - \varphi_\lambda^{j,-i,-j})\Big] + \mathbb{E}\Big[(\varphi_\lambda^i - \varphi_\lambda^{i,-i,-j})\varphi_\lambda^{j,-i,-j}\Big] \right|
$$

$$
\leq \mathbb{E}\Big[|\varphi_\lambda^i||\varphi_\lambda^j - \varphi_\lambda^{j,-i,-j}|\Big] + \mathbb{E}\Big[|\varphi_\lambda^{j,-i,-j}||\varphi_\lambda^i - \varphi_\lambda^{i,-i,-j}|\Big] \leq \frac{64\big\|\frac{\delta^2 \varphi}{\delta m^2}\big\|_\infty \big\|\frac{\delta \varphi}{\delta m}\big\|_\infty}{N}.
$$

Therefore, we have

$$
\mathbb{E}\big[S_1^2\big] \leq \frac{16\big\|\frac{\delta \varphi}{\delta m}\big\|_\infty^2}{N} + \frac{128\big\|\frac{\delta^2 \varphi}{\delta m^2}\big\|_\infty \big\|\frac{\delta \varphi}{\delta m}\big\|_\infty}{N},
$$

and thus

$$
\mathbb{E}\Big[\big|\varphi(m_T) - \varphi(m_T^N)\big|^2\Big] = \mathbb{E}\big[|S_1 + S_2|^2\big] \leq \frac{C}{N},
$$

where $C$ depends only on the bounds on the first and second-order derivatives. $\qquad\square$

## B.3 Proof of Lemma 7.1

*Proof of Lemma 7.1.* The bound of $\|u^{(k)}\|$ and $\|v^{(k)}\|$ can easily be obtained by its probability representation. As for the gradients' bound, For $t \in [t_k, t_{k+1})$, given $\overleftarrow{X}_{t_k}^h$, $\overleftarrow{X}_t^h$ follows an OU process, then we can write

$$
\overleftarrow{X}_{t_{k+1}}^h \sim \mathrm{N}\left( \overleftarrow{X}_t^h e^{-g(T-t)+g(T-t_{k+1})} + S_{T-t_k}^*(\overleftarrow{X}_{t_k}^h)\big(1 - e^{-g(T-t)+g(T-t_{k+1})}\big), \frac{1 - e^{-2(g(T-t)-g(T-t_{k+1}))}}{2} \right).
$$

Then,

$$
u^{(k)}(t,x;x_k) = \mathbb{E}\Big[v^{(k+1)}(\overleftarrow{X}_{t_{k+1}}^h)\Big|\overleftarrow{X}_t^h = x, \overleftarrow{X}_{t_k}^h = x_k\Big] = \mathbb{E}\Big[v^{(k+1)}\big(\sigma^{(k)}(t)Z + \mu^{(k)}(t,x;x_k)\big)\Big],
$$

and

$$
v^{(k)}(x) = \mathbb{E}\Big[v^{(k+1)}(\overleftarrow{X}_{t_{k+1}}^h)\Big|\overleftarrow{X}_{t_k}^h = x\Big] = \mathbb{E}\Big[v^{(k+1)}\big(\sigma^{(k)}(t_k)Z + \mu^{(k)}(t_k,x;x)\big)\Big],
$$

where $Z \sim \mathrm{N}(0, I_d)$ is standard normal and

$$
(\sigma^{(k)})^2(t) := \frac{1 - e^{-2(g(T-t)-g(T-t_{k+1}))}}{2}, \quad \mu^{(k)}(t,x;x_k) = xe^{-g(T-t)+g(T-t_{k+1})} + S_{T-t_k}^*(x_k)\big(1 - e^{-g(T-t)+g(T-t_{k+1})}\big)
$$

Then,

$$
\partial_{x_i} v^{(k)}(x) = \mathbb{E}\left[ \frac{\mathrm{d}\, v^{(k+1)}\big(\sigma^{(k)}(t_k)Z + \mu^{(k)}(t_k,x;x)\big)}{\mathrm{d}x_i} \right]
$$

$$
= \mathbb{E}\Bigg[ \sum_{j=1}^d \big(1 - e^{-g(T-t_k)+g(T-t_{k+1})}\big)\partial_{x_i}(S_{T-t_k}^*(x))_j \cdot \partial_{x_j} v^{(k+1)}\big(\sigma^{(k)}(t_k)Z + \mu^{(k)}(t_k,x;x)\big)
$$

$$
+ e^{-g(T-t_k)+g(T-t_{k+1})}\partial_{x_i} v^{(k+1)}\big(\sigma^{(k)}(t_k)Z + \mu^{(k)}(t_k,x;x)\big) \Bigg],
$$

which leads to

$$
\|\nabla v^{(k)}\|_\infty \leq \|\nabla v^{(k+1)}\|_\infty \cdot \Big( e^{-g(T-t_k)+g(T-t_{k+1})} + \big(1 - e^{-g(T-t_k)+g(T-t_{k+1})}\big)\|S^*\|_{\mathrm{Lip},1,\infty} \Big). \tag{20}
$$

By iteration, we get

$$
\begin{aligned}
\left\|\nabla v^{(k)}\right\|_\infty &\leq \left\|\nabla v^{(K)}\right\|_\infty e^{\sum_{\ell=k}^{K-1}(g(T-t_{k+1})-g(T-t_k))} \prod_{\ell=k}^{K-1}\left(1+\left(e^{g(T-t_\ell)-g(T-t_{\ell+1})}-1\right)\|S^*\|_{\mathrm{Lip}}\right) \\
&= \|\nabla\varphi\|_\infty \exp\left\{g(T-t_K)-g(T-t_k)+\sum_{\ell=k}^{K-1}\ln\left(1+\left(e^{g(T-t_\ell)-g(T-t_{\ell+1})}-1\right)\|S^*\|_{\mathrm{Lip},1,\infty}\right)\right\} \\
&\leq \|\nabla\varphi\|_\infty \exp\left\{g(T-t_K)-g(T-t_k)+\|S^*\|_{\mathrm{Lip},1,\infty}\sum_{\ell=k}^{K-1}\left(1-e^{g(T-t_\ell)-g(T-t_{\ell+1})}\right)\right\} \\
&\leq \|\nabla\varphi\|_\infty \exp\left\{g(T-t_k)-g(T-t_K)+\|S^*\|_{\mathrm{Lip},1,\infty}\sum_{\ell=k}^{K-1}g(T-t_\ell)-g(T-t_{\ell+1})\right\} \\
&\leq \|\nabla\varphi\|_\infty \exp\left\{\left(g(T-t_k)-g(T-t_K)\right)\left(1+\|S^*\|_{\mathrm{Lip},1,\infty}\right)\right\},
\end{aligned}
\tag{21}
$$

which leads to the desired bound of $\|\nabla v^{(k)}\|_\infty$. Using the similar argument in equation 20 , we can also get for all $y\in\mathbb{R}^d$,

$$
\begin{aligned}
\partial_{x_i}u^{(k)}(t,x;y) &= \mathbb{E}\left[\frac{\mathrm{d}\,v^{(k+1)}\left(\sigma^{(k)}(t)Z+\mu^{(k)}(t,x;y)\right)}{\mathrm{d}x_i}\right] \\
&= \mathbb{E}\left[e^{g(T-t)-g(T-t_{k+1})}\partial_{x_i}v^{(k+1)}\left(\sigma^{(k)}(t)Z+\mu^{(k)}(t_k,x;x_k)\right)\right],
\end{aligned}
$$

which means for all $y\in\mathbb{R}^d$,

$$
\left\|\nabla u^{(k)}(t,\cdot;y)\right\|_\infty \leq e^{g(T-t)-g(T-t_{k+1})}\left\|\nabla v^{(k+1)}\right\|_\infty.
$$

Combining with equation 21, we have for all $y\in\mathbb{R}^d$.

$$
\left\|\nabla u^{(k)}(t,\cdot;y)\right\|_\infty \leq \|\nabla\varphi\|_\infty e^{(1+\|S^*\|_{\mathrm{Lip},1,\infty})g(T)}.
\tag{22}
$$

which completes the proof of the lemma. $\qquad\square$

