# OpenReview forum: "Dimension-free error estimate for diffusion model and optimal scheduling"
_TMLR — Accepted by TMLR_

### Review · Reviewer_S1gY · 2026-01-05

**Summary Of Contributions:**

## Contributions

This paper studies the convergence of score-based generative models under a novel criterion: the expected distributional error on a class of smooth functionals on the space of probability distributions. This is a weaker condition then the convergence in terms of KL or W2, and the authors obtained dimension-free convergence rate under finite training data samples.

The key technical tools to establish the convergence of continuous-time dynamics are from mean-field SDEs (which I haven't fully understood due to my limited background in this area), and to analyze the discretization error, the additional tool is from the renowned Conforti et al. 2025 paper. The authors also provided insights into the optimal choice of schedulers that minimizes an upper bound on the discretization error, which is the first rigorous study of this issue to the best of our knowledge. This optimal scheduler turns out to be tight in the Gaussian case, which is a nice property. (For discrete diffusion model, https://arxiv.org/abs/2508.04884 showed that the cosine schedule is optimal under Fisher-Rao geometry. I'm curious if your obtained scheduler is also Fisher-Rao optimal).

Overall, I think this is a solid theoretical contribution to the understanding of score-based generative models, and I recommend acceptance at TMLR.

## Weaknesses

The main weakness of this paper is that the authors haven't explained the criterion they used to measure the distributional error in detail. As discussed above, this is a weaker condition than the commonly used metrics, and the authors haven't discussed its connections to other metrics and its implications in practice. For instance, the metric $G(m)=\langle\varphi,m\rangle$ for some smooth test function $\varphi$ should be connected to the bounded Lipschitz distance if you assume $\varphi$ has regularity up to the first order. Are there any other examples of $G$ beyond inner products with smooth functions? Also, if the authors can provide some examples showing such metric can be much weaker than W2 or TV distance (e.g., when TV is large while the error in $G$ converges to zero), that would be helpful for readers to understand the implications of this metric.

**Additional Comments:**

- Do the authors think the same approach can be used for analyzing the convergence of *equilibrium* MCMC samplers such as Langevin dynamics / Langevin Monte Carlo / proximal sampler?

**Audience:**

Yes

**Audience Explanation:**

I myself feel the paper has provided an interesting perspective to analyze the convergence of diffusion model, and I believe it would be interesting to people in sampling theory and probabilistic methods.

**Claims And Evidence:**

Yes

**Claims Explanation:**

The claims made in the paper are supported by well-written proofs and experimental results.

**Requested Changes:**

- Could the authors provide some examples of the functional $G$ and how to compute their linear derivative?
- In theorem 3.5, the authors haven't made it clear the constant $C$ and $C_T$. How do they depend on the bound of derivatives in assumption 3.4? The current statement seems to suggest that when these derivatives are all $O(1)$ then $C,C_T$ are also $O(1)$, but what about the case when these derivatives scale with dimension? Also, I suggest the authors to provide some brief discussion of the obtained bound in both theorem 3.5 and 3.10, and in particular what each term represents, although it seems obvious to readers familiar in the analysis of diffusion models.

Minor comments:

- This paper uses $m$ to denote the probability distributions, which is not a common practice in the literature and I suggest the authors to use more standard notation such as $p$ or $\mu$. Also, the notation $\nabla\log p$ in the discrete-time setting is originally $\nabla\log\widetilde{p}$ in the Conforti et al. 2025 paper.
- Paragraph before Eq (1): $\mathrm{Law}(\overleftarrow{X}_0)=m_T^N$.
- In assumption 3.1, the meaning of $H$ is not introduced.

---

> ### Author Response · Authors · 2026-02-09
> **Response to the Reviewer**
>
> We thank the Editor and the Reviewers for their careful reading of our manuscript and for their constructive comments. We believe that these suggestions have significantly improved the quality and clarity of the paper. Below, we address each comment point by point. All changes made in the revised manuscript in response to the Reviewers’ suggestions are highlighted in blue. Due to the character limit, we present our response in several comments.
>
> **Response to requested changes.**
> 1. We added clarifying examples of the linear derivative (see Remark 3.4 of the revised version). For instance, if $G(m) = \int f(x) m(dx)$,
> then $\frac{\delta G}{\delta m} = f(x)$. For a cylindrical functional of the form
> $$G(m)= F\Bigl(\int f_1(x) m(dx), \int f_2(x) m(dx),.., \int f_k(x) m(dx)\Bigr)$$
> we have
> $$\frac{\delta G}{\delta m} = \sum_{i=1}^k\partial_iF\Bigl(\int f_1(x) m(dx), \int f_2(x) m(dx),.., \int f_k(x) m(dx)\Bigr)f_i(x).$$
>
> 2. Regarding the constants $C$ and $C_T$:
> a) As shown in Sections 6 and 7 and Appendix B, these constants are proportional to the square of the bounds on the intrinsic and linear derivatives. This relationship is clarified in the revision.
> b) Our analysis assumes these derivative bounds are dimension-independent. We added a remark explicitly connecting $C$ and $C_T$ to these bounds, noting that the constants may become dimension-dependent if the derivatives themselves are (see Theorem 3.7 and Remark 3.11 in the revised version).
> c) We included a brief discussion on the interpretation of each term in Theorems 3.5 and 3.10 (Theorems 3.7 and 3.16 in the revised version), see Remark 3.12.

---

> > ### Author Response · Authors · 2026-02-09
> > **Response to the Reviewer**
> >
> > **Response to minor comments.**
> > Thank you for the suggestions.
> > 1. The use of notation $m$ for probability measure is rather standard, see e.g. [1].
> > 2. ﻿﻿﻿The typographical error was corrected in the revised version.
> > 3. The notation H was introduced in Assumption 3.1 of the revised version.
> >
> > **Response to additional comments.**
> > We appreciate the insightful suggestion. We believe our methodological framework could be extended to analyze the convergence of equilibrium MCMC samplers, such as Langevin dynamics.
> >
> > **Response to questions raised in Contributions and Weaknesses section.**
> > 1. We thank the Reviewer for pointing us to the paper on the optimality of a scheduler— formulated in terms of the Fisher–Rao metric—for the masked DDPM model. Our notion of an optimal schedule is indeed closely related at an intuitive level.
> > More precisely, consider the time-rescaled Ornstein--Uhlenbeck process
> > $$
> > dX_t = -\dot g(t) X_t dt + \sqrt{\dot g(t)} dW_t . \quad (1)
> > $$
> > One can show that the relative Fisher information satisfies the sharp decay estimate
> > $$
> > I(m_t \mid m^\*) \approx e^{-2g(t)} I(m_0\mid m^\*),
> > $$
> > where $m^\*$ denotes the invariant (Gaussian) measure of the OU dynamics. Moreover, the relative entropy $H(m_t \mid m^\*)$ satisfies the energy dissipation identity
> > $$
> > dH(m_t\mid m^\*) = - I(m_t\mid m^\*) \dot g(t) dt.
> > $$
> > In our setting, the proposed optimal scheduler $g^*$ is obtained by minimizing the functional
> > $$
> > \int_0^T e^{-2g(t)} |\dot g(t)|^2 dt,
> > \qquad \text{subject to } g(T)=T'.
> > $$
> > This choice yields a scheduler that equalizes, in time, the effective ``loss of energy'' along the evolution over $[0,T]$. In this sense, our criterion can be viewed as an analogue of the approach in the cited paper: while the authors characterize an optimal scheduler by matching a geodesic under the Fisher–Rao metric, we instead select a scheduler that enforces a uniform dissipation rate of the entropy/energy over time.
> >
> > If the initial law for (1) is $m_0 = \mathcal{N}\Bigl( \mu, \frac12  \Bigr) $ with the same variance as the invariant measure, then it can be shown that the scheduler $ g^* $ is also optimal in the sense of Fischer–Rao metrics, i.e. it gives a constant-speed Fischer–Rao geodesics. In general, it is not true. See Remark 3.19 of the revised version.
> >
> > 2. The metrics is weaker than the Wasserstein metrics in the following sense.
> > When measuring the distance between two measures in the Wasserstein-1 metrics, the difference between the measures is tested against Lipschitz functions with Lipschitz constant smaller than 1:
> > $$
> > W\_1(\mu, \nu) = \sup_{\\| f \\|\_{Lip}  \leq 1 } \langle f, \mu - \nu \rangle.
> > $$
> > Note that $W_2$ is stronger, i.e. $W_1( \mu, \nu ) \leq W_2(\mu, \nu )$ .
> > For metrics defined via G the testing is performed against a narrower class of regular test functions. To measure the TV distance between two functions one needs to consider even a larger set of test functions, namely, all bounded measurable functions:
> > $
> > \\|\mu- \nu \\|\_{TV} = \frac{1}{2} \sup\_{\\|f \\|\_{\infty} \leq 1 } \langle f,\mu - \nu \rangle
> > $.
> >
> > The differences between the metrics considered above leads to the following comparison between continuous measure $\mu$ with its empirical counterpart $\mu^N$ in terms of these metrics. Note, that due to the classical result of [2] we have $ W_1(\mu, \mu^N ) = O(1 / N^{1/d} ) $. We also have that $\\|\mu - \mu^N \\|_{TV} = 1 $ (does not decrease as N augments). At the same time, with the metrics considered in the present paper we get $ G(\mu) - G(\mu^N) = O( 1/ \sqrt{N} ) $ (see Lemma 6.2).
> >
> > An example of a metric related to a functional $G$ satisfying the assumptions of the paper is the MMD metrics:
> > $$\mathrm{MMD}\_k^2(\mu,\nu) := \iint k(x,x')  d\mu(x) d\mu(x') + \iint k(y,y')  d\nu(y) d\nu(y') - 2 \iint k(x,y)  d\mu(x) d\nu(y).$$
> > If $k$ is, for example, a Gaussian kernel (often used in practice), then $G:\mu\mapsto {\rm MMD}_k^2(\mu,\nu)$ admits bounded 1st and 2nd order linear and intrinsic derivatives.
> >
> > This comment was added as Remark 3.10 in the revised version.
> >
> > We thank the Reviewer for the valuable feedback.
> > We hope that the revision satisfactorily addresses all concerns.
> >
> > **References**
> >
> > [1] Pierre Cardaliaguet, Francois Delarue, Jean-Michel Lasry, and Pierre-Louis Lions. The master equation and the convergence problem in mean field games. In The Master Equation and the Convergence Problem in Mean Field Games, volume 201 of Annals of Mathematics Studies. Princeton University Press, Princeton, NJ, 2019.
> >
> > [2] Nicolas Fournier and Arnaud Guillin. On the rate of convergence in Wasserstein distance of the empirical measure. Probab. Theory Relat. Fields, 162(3–4):707–738, 2015.

---

### Review · Reviewer_g4b7 · 2026-01-17

**Summary Of Contributions:**

This paper provides dimension-free weak error estimates for diffusion-model generation and uses them to derive an explicit “optimal” time schedule for discretized sampling in an OU/VP setting.

Main contributions:\
1.	Weak functional metric for generative error. Instead of KL (ill-posed for empirical measures) or Wasserstein (dimension-cursed for empirical convergence), the paper measures discrepancy via smooth distribution functionals $G:\mathcal P_2(\mathbb R^d)\to\mathbb R$ with bounded derivatives, and bounds $\mathbb E|G(\mu_{T-\delta})-G(m_0)|^2$. This choice enables statistical 1/N-type rates without explicit d dependence in the stated metric.\
2.	Continuous-time end-to-end error decomposition. Theorem 3.5 gives a bound decomposing error into (i) finite-sample $(\sim 1/N)$, (ii) finite mixing $(\sim e^{-2\rho T})$, (iii) score approximation (an integrated $\varepsilon^2$ term), and (iv) a small-time cutoff penalty $\delta$ to avoid singularities at t=0.\
3.	Discrete-time weak error + schedule optimization. For discretized reverse OU and linear observables $G(m)=\langle\phi,m\rangle$, the paper derives a schedule-dependent discretization bound (Thm. 3.10) and optimizes an approximation to obtain a closed-form schedule which matches commonly used schedules and is related to flow matching/rectified flow parameterizations.\
4.	Brief discrete-state illustration of the same weak-metric philosophy when KL is problematic for sparse empirical distributions.

Key strengths: \
1.	clear motivation for the weak metric\
2.	interpretable decomposition of error sources\
3.	explicit schedule prescription that plausibly justifies a popular heuristic.

Key weaknesses: \
1.	dimension-free hinges on strong smoothness assumptions on G (potentially missing perceptual/high-frequency errors and hiding d in constants)\
2.	schedule optimality is for a specific proxy bound under restrictive settings (OU/linear observables)\
3.	limited empirical validation and unclear interaction with score training (since changing g changes the training time/noise distribution and thus $\varepsilon$).

**Audience:**

Yes

**Audience Explanation:**

Yes. The paper should interest a meaningful subset of TMLR readers, especially those working on (i) theoretical foundations of diffusion-based generative models, and (ii) principled design of time schedules.

In particular, the paper’s value to this audience is:
1) A “dimension-free” perspective via weak metrics: It reframes convergence/error analysis away from KL/Wasserstein toward smooth test-functionals where empirical convergence can avoid explicit dimension dependence, which is conceptually relevant for high-dimensional generative modeling.
2) An interpretable error decomposition: The continuous-time bound cleanly separates finite-sample, mixing, score approximation, and cutoff effects, which is useful for reasoning about where generation error comes from in theory.
3) Scheduling as optimization of discretization bias: The derivation of an explicit schedule aligned with commonly used heuristics provides a potentially useful theoretical justification and could motivate more principled schedule choices.

To summarize, the audience interest will be mostly on the theory side and not on the applied side, unless the paper strengthens empirical validation and clarifies how the weak-metric guarantees relate to standard evaluation criteria and to training–schedule interactions.

**Broader Impact Concerns:**

I do not see major novel ethical concerns specific to this submission beyond those typical for work that can improve generative modeling (e.g., potential downstream misuse for generating deceptive or harmful content).

**Claims And Evidence:**

Yes

**Claims Explanation:**

The core theoretical claims are largely supported within the paper’s stated setting and weak metric, but several interpretations (dimension-free, optimal scheduling, and the flow-matching connection) are not yet backed by sufficiently clear scoping or empirical evidence.

Supported:\
i) The continuous-time weak error result and its decomposition (finite-sample, mixing, score error, cutoff $\delta$) are presented coherently under explicit assumptions, and the derivations appear consistent with standard weak-error tools.\
ii) The discrete-time analysis yields a schedule-dependent discretization bound and the derivation of the closed-form $g^\*$ is convincing as an optimizer of the chosen approximate bound in the OU setting.

Under-supported:\
iii) “Dimension-free”: while d does not appear explicitly, dependence can re-enter through derivative norms/regularity constants (and Fisher-information-like terms). The paper does not clearly characterize when these remain dimension-stable for practically relevant observables.\
iv) “Optimal schedule” as an end-to-end prescription: optimizing discretization alone ignores that changing g(t) changes the training time/noise distribution and thus the score error $\varepsilon$ (likely $\varepsilon=\varepsilon(g)$). Without theory or ablations with matched training reweighting, “optimal” is only partially justified.\
v) Empirics are too limited to support broad practical claims about scheduling benefits with learned scores.\
vi) The flow-matching/rectified-flow connection is plausible but asserted more than demonstrated. The explicit time-variable mapping and the precise sense of equivalence should be spelled out.


Overall: Solid theoretical evidence for the stated weak-metric OU results with weaker support for the broader practical and interpretive claims.

**Requested Changes:**

Major:

1) Scope “dimension-free” precisely:
Make explicit what constants depend on (e.g derivative norms, information terms) and discuss how they may scale with d. Provide at least one concrete, nontrivial class of observables G for which the required bounds remain dimension-stable (e.g., low-dimensional feature/projection expectations), and note what discrepancies the weak metric may miss (high-frequency/perceptual artifacts).
2) Address schedule–training interaction ($\varepsilon$ likely depends on g):
Either include theory/discussion acknowledging $\varepsilon=\varepsilon(g)$, or add an empirical ablation: baseline vs $g^\*$ with unchanged training time sampling/weighting vs $g^\*$ with matched reweighting sampling.
3) Strengthen empirics beyond 1D toy: Add at least one higher-dimensional controlled experiment (e.g., known-score synthetic or learned score in moderate d) validating step-size scaling and schedule effects, ideally also reporting whether gains transfer to at least one standard metric.

Minor:

4) Make the flow-matching / rectified-flow connection explicit. Add the exact mapping between OU time s, algorithmic time t via s=g(t), and FM/RF interpolation $\tau\in[0,1]$; clarify in what sense the schedules coincide (e.g., linearizing $\alpha=e^{-s}$ in $\tau$, not full equivalence of coupling paths of VP spherical interpolant vs linear RF path).
5) Discuss proxy-optimality vs practical optimality. How tight is the bound and how sensitive is performance to deviations from $g^\*$?
6) Guidance for $\delta, T, T'$. Provide heuristics for choosing the cutoff $\delta$ and time horizons balancing mixing vs score accumulation.
7) Discrete state-space section: deepen or streamline. Currently, it reads as mainly illustrative. Clarify whether it is intended as conceptual motivation or a true extension. If it is an extension, add at least one nontrivial discrete example (e.g., larger state space, and whether any schedule implications carry over). Otherwise, consider moving it to an appendix.

---

> ### Author Response · Authors · 2026-02-09
> **Response to the Reviewer**
>
> We thank the Editor and the Reviewers for their careful reading of our manuscript and for their constructive comments. We believe that these suggestions have significantly improved the quality and clarity of the paper. Below, we address each comment point by point. All changes made in the revised manuscript in response to the Reviewers’ suggestions are highlighted in blue. Due to the character limit, we present our response in several comments.
>
> **Response to requested changes**
> 1. In Theorems 3.5 and 3.10 (Theorems 3.7 and 3.16 in the revised version) we have explicited the dependence of the constants on the derivatives of the functional, the information term and $T$.
> Let $r < d$ and let $A\in\mathbb R^{r\times d}$ satisfy$\\|A\\|\le 1$.
> For any $\phi\in C_b^2(\mathbb R^r)$, define
> $$
> G(m):=\int\_{\mathbb R^d} \phi(Ax) m(dx).
> $$
> This functional admits bounded first and second linear and intrinsic derivatives with bounds not depending on the ambient dimension $d$. Since the metric only probes $m$ through test functions defined on a low-dimensional feature space, it can be insensitive to high-frequency differences in the ambient space. Concretely, two distributions can agree on low-complexity observables while differing on fine-scale structure (e.g. narrow spiky components, high-frequency texture, or perceptual artefacts).
> This comment was added as Remark 3.6 in the revised version.
>
> 2. The Assumption 3.13 of paper states that we assume that for given $ \epsilon $ and $ g $ there exists a score-matching function $ S^* $ such that the property (3) holds, i.e. $ S^* = S^\*( \epsilon, g ) $ (we added a comment after Assumption 3.13). The score function $ S^\* $ only contributes to the error bound through the Lipschitz constant $L$ which is assumed to be independent of $g$. This assumption is reasonable due to the fact that the real score function is Lipschitz; see Remark 3.3.
> 3. We added a higher-dimensional controlled experiment (known-score for Gaussian mixture in $ \mathbb{R}^2 $). The results are compared in terms of mean log-likelihood, Wasserstein W1, Wasserstein W2 and MMD metrics (see end of Section 4 in the revised version).
>
> 4. FM/RF connection via time change.
> Consider the scheduled OU forward SDE
> $$
> dX_t = -\dot g(t) X_t dt + \sqrt{\dot g(t)} dW_t,\qquad t\in[0,T_{\mathrm{alg}}],
> $$
> where $g:[0,T_{\mathrm{alg}}]\to[0,T']$ is an increasing scheduler.
> With the time change $s = g(t)$
> this is exactly a time-rescaled version of the standard Ornstein–Uhlenbeck process
> $$
> dX_s = -X_s ds + dW_s.
> $$
> Consequently, the forward marginal admits the closed-form coupling
> $$
> X_t \equiv X_{s=g(t)} = \alpha(t) X_0 + \sigma(t) Z,\qquad Z\sim\mathcal N(0,I),\ Z\perp X_0,
> $$
> with coefficients $\alpha(t)=e^{-g(t)}$, $\sigma(t)=\sqrt{1-e^{-2g(t)}}$.
> Defining the interpolation parameter
> $$
> \tau_{\mathrm{VP}} := \sigma(t)=\sqrt{1-e^{-2g(t)}}\in[0,1),
> $$
> we obtain
> $$
> X_t = \sqrt{1-\tau_{\mathrm{VP}}^{2}}X_0 + \tau_{\mathrm{VP}}Z,
> $$
> which is precisely the spherical (variance-preserving) interpolant used in
> flow-matching for VP diffusions.
> Rectified flow instead uses the linear coupling
> $$
> \widetilde X_\tau = (1-\tau)X_0 + \tau Z,\qquad \tau\in[0,1].
> $$
>
> If we define
> $$
> \tau_{\mathrm{RF}} := 1-\alpha(t)=1-e^{-g(t)},
> $$
> then the data coefficient is linearized, $ \alpha(t)=1-\tau_{\mathrm{RF}} $,
> but the corresponding noise coefficient becomes
> $
> \sigma(t)=\sqrt{1-(1-\tau_{\mathrm{RF}})^2}
>        =\sqrt{\tau_{\mathrm{RF}}(2-\tau_{\mathrm{RF}})},
> $
> which differs from $\tau\_{\mathrm{RF}}$. Therefore, this mapping matches RF only
> at the level of the $\alpha$-schedule, but not the full linear coupling path.
> Under the optimal scheduler $g^\*(t)$ defined in the paper
> $$
> \alpha^\*(t)=e^{-g^\*(t)} 1 - t \frac{ 1 - e^{-T'} }{ T_{alg}}
> $$
> is affine in $t$, i.e. $\tau_{\mathrm{RF}}=1-\alpha^\*(t)$ is linear in algorithmic
> time. This is the precise sense in which our scheduler aligns with rectified-flow
> parameterisations. However, the resulting coupling path remains the VP/spherical
> interpolant, not the linear RF path.
> The remark on the connection with VP/RF flow-matching was added as Remark 4.1 in the revised version.

---

> ### Author Response · Authors · 2026-02-09
> **Response to the Reviewer**
>
> 5. Let $g^\*\_t$ be the optimal scheduler and $\mathcal{G}(g_t) = \int\limits_0^T | \dot g_t |^2 e^{ - 2 g_t } dt$. Consider a small variation $ \delta_t $ of $g_t^\*$ verifying $  \delta_t(0) = 0, \delta_t(T)=0$. Then
> $$\mathcal{G}(g_t^\* + \delta_t) - \mathcal{G}( g^\*_t ) = e^{-C_2}\int_0^T (C_1 - t)^2 \dot\delta_t^2 dt + o( \delta_t^2 ),$$
> where constants $C_1$, $C_2$ are defined in the proof of Corollary 3.18. This term gives a measure of sensitivity of the error with respect to deviations from $g^\*$.
> The corresponding remark has been added after the proof of Corollary 3.18.
> 6. In the continuous time setting, given a desired error level $\mathrm{err}$, one can take $ \delta \leq \sqrt{ \frac{\mathrm{err}}{C_4} } $, $ N \geq \frac{C_1}{\mathrm{err}} $, and then choose $ T $ of order $ \frac{1}{2\rho} \log\left( \frac{\mathrm{err}}{C_2} \right) $ and the score-matching error of order $ \frac{1}{\alpha^2 + 1}\sqrt{ \frac{\mathrm{err}}{C_3} } $, where
> $ C_i $, $ i = 1, 2, 3, 4 $, are defined in Theorem 3.7. The corresponding remark was added after the Theorem 3.7 in the revised version (see Remark 3.12).
> In the discrete time setting one can take $ T = 1 $; then $T' = g(1)$ plays the role of $T$ of the continuous time setting.
>
> 7. Thank your for this suggestion to improve the streamline of the paper. This section was moved to appendix.
>
> We thank the Reviewer for the valuable feedback.
> We hope that the revision satisfactorily addresses all concerns.

---

### Review · Reviewer_2Vhg · 2026-01-30

**Summary Of Contributions:**

This paper presents a dimension-free error estimate for sampling with diffusion models. The guarantees are formulated in both the continue time reverse process and the discrete time reverse process, in terms of a “smooth” functional G defined over probability measures, equal to zero at the target.


Strengths:

The approach relies on backward Kolgmorov in a space of probability measures and is rather elegant

Weaknesses:

Clarity. The approach is clean, but at the cost of losing the reader. The paper employs tools from Lions calculus etc which are not standard to the average TMLR reader. A bit of context is needed for me to check the correctness of the equations

Smooth functional. The authors should spend time trying to instantiate the functional. For example, does it exist G such that G(mu) = G(m) implies mu = mu? With that separation property, the authors might derive (asymptotic) convergence from their bounds.

Dimension-free. It’s a bit over claiming, I would say that the bounds do not depend explicitly on the dimension.

**Audience:**

Yes

**Audience Explanation:**

Theory for sampling with diffusion models is a relevant topic for TMLR

**Claims And Evidence:**

No

**Claims Explanation:**

I could not understand parts of the paper, specifically:


Assumption 3.1: H is not defined
Assumption 3.2. 1: maybe recall that the true score is Lipschitz on [delta, T]
Theorem 3.5.: Is N superscript missing in the l.h.s? Asking because there is N in the r.h.s.
Remark 3.8 The linear part in the score (that comes from the Gaussian) could be integrated in closed form no? Why do you learn it?
Theorem 3.10: Does it apply only for functionals G linear in m (G(m) = <phi,m>) ?
Remark 3.12: typo in the first equation g_{T-t} vs g(T-t). Besides, I had hard time understanding this remark, please clarify.
Section 4: I understand it is a restricted setting, but I am wondering, do you claim the last equation before remark 4.1 (with a formal proof), or can you rigorously prove it? The status of this inequality should be clarified “claim”/“formal proof”/“rigorous proof possible but not in included in the paper”
Section 5: My understanding is that the scheduler is only “nearly” the optimal one. So “nearly optimal” is more accurate than “optimal”
Proof of Theorem 3.5 (also applies to Proof of Theorem 3.10). Some material recalling how Backward Kolmogorov works in space of probability measures would be appreciated
“Note that by the flow property…” I think the statement is trivial but difficult to parse because of the notation. The authors could clarify to help the reader
First equation where H appears: What is the constant C here (in front)?
Second and third = symbol: In general I was not familiar with Backward Kolmogorov in the space of probability measures. Did you use linearity for D_m U ?

**Requested Changes:**

Just clarify the points above please.

Besides I was wondering:
Do you measure distance to m_0^N? Where does the need to use m_0^N come from?

---

> ### Author Response · Authors · 2026-02-09
> **Response to the Reviewer**
>
> We thank the Editor and the Reviewers for their careful reading of our manuscript and for their constructive comments.
> We believe that these suggestions have significantly improved the quality and clarity of the paper.
> Below, we address each comment point by point. All changes made in the revised manuscript in response to the Reviewers’ suggestions are highlighted in blue. Due to the character limit, we present our response in several comments.
>
> **Response to questions raised by the Reviewer**
> 1. Assumption 3.1: the definition of $H$ was added to Assumption 3.1.
> 2. Assumption 3.2.1: Remark 3.3 contains a comment on the Lipschitz property of the empirical score function $ \nabla \log m^N_t $.
> 3. Theorem 3.5 (Theorem 3.7 in the revised version): the dependence on $N$ in the l.h.s. of the expression in Theorem 3.5  is present implicitly: note that $ \mu^\*\_{T-\delta} $ is the law of the time-reversed diffusion process with the learned score function $ S^*_{T-t} $ which is chosen to match the empirical score $ m^N_{T-t} $ (see Assumption 3.2). A comment was added to the Remark 3.8.
> 4. Remark 3.8 (Remark 3.14 in the revised version): it is true that $ \nabla \log p^N_t $ and $ \nabla \log m^N_t $ only differ by a linear term; for the sake of mathematical analysis we chose to study $ \nabla \log p^N_t $ rather than $ \nabla \log m^N_t $ since $ \nabla \log p^N_t \to 0 $ as $ t \to \infty $ exponentially quickly, and that is what allows to obtain the discrete-time error bound and the optimal scheduler.
> 5. Theorem 3.10 (Theorem 3.16 in the revised version): indeed, Theorem 3.16 is proved only for linear functionals. We have precised in the formulation of the theorem that $G$ is taken to be of the form $ G( m ) = \langle \phi, m \rangle $
> 6. Remark 3.12 (Remark 3.17 in the revised version): thank you, the typo was corrected in the revised version. The remark was reformulated to improve clarity.
> 7. Section 4 (Section A of the Appendix in the revised version): upon the suggestion of another Reviewer this section was moved to Appendix; it contains sketches of proofs of the claimed results. Thus a rigorous proof of the result preceding Remark A.1 of the revised version is possible, but not included in the paper.
> 7. Section 5 (Section 4 of the revised version): the scheduler is indeed nearly optimal in the sense precised in the Remark 3.17 of the revised version.
> 8. Proof of Theorems 3.5 and 3.10 (Theorems 3.7 and 3.16 in the revised version): We added the reference to Theorem 7.2 in [1] for the readers interested in the Backward Kolmogorov equations.
> 9. ``Note that by the flow property...'': by definition of $ U $ we have trivially that $ U( T - \delta, \mu^\*\_{ T - \delta } ) = G\big({\rm Law}(\overleftarrow{X}^\*\_{T-\delta}|\overleftarrow{X}^\*\_{T - \delta}\sim \mu^\*\_{T-\delta})\big) = G( \mu^\*\_{ T - \delta } ) $. Also $ U( 0, \mu^\*\_{ 0 } ) = G\big({\rm Law}(\overleftarrow{X}^\*\_{T-\delta}|\overleftarrow{X}^\*\_{0}\sim m^\*)\big) = G( \mu^\*\_{ T - \delta } ) $ by the definition of the time-reversed diffusion with score-matching function in Section 3. We added a clarification to the proof of Theorem 3.7 in the revised version.
> 10. ``First equation where H appears'': the constant in front of $H$ is $\frac12$ and comes from Pinsker inequality: $ \\|\mu - \nu\\|^2_{TV} \leq \frac{1}{2} H( \mu \mid \nu ) $, where $ \\| \mu - \nu \\|_{TV} $ denotes the total variation distance between $ \mu $ and $ \nu $. We have added an extra explanation concerning the use of this inequality in the Theorem 3.7 in the revised version.
> 11. Second and third = symbol: The second equation holds due to the Ito's formula for measure processes (see for instance Theorem 5.99 of [2]) and the third equation holds due to the backward Kolmogorov equation. We have added this explanation to the proof of Theorem 3.7 in the revised version.
>
> **References**
>
> [1] Rainer Buckdahn, Juan Li, Shige Peng, and Catherine Rainer. Mean-field stochastic differential equations and associated PDEs. The Annals of Probability, 45(2):824 – 878, 2017.
>
> [2] Rene Carmona and Francois Delarue. Probabilistic theory of mean field games with applications I-II, volume 3. Springer, 2018.

---

> > ### Author Response · Authors · 2026-02-09
> > **Response to the Reviewer**
> >
> > **Response to questions raised in Weaknesses section**
> > 1. Several remarks were added to the revised version to clarify the use of Lions calculus, see in particular Remark 3.4 for some examples of calculation of the linear derivative, Remark 3.6 for an example of a functional $G$ for which the linear and intrinsic derivatives do not depend on the dimension $d$ and Remarks 3.10 and 3.11 for an example of a functional $G$ satisfying the assumptions of the paper and frequently used in practice (MMD).
> > 2. A functional of the type $ G( m ) = \langle \phi, m \rangle $ for a given $ \phi $ does not separate measures. However, one can take a supremum over all the $ C^2 $ bounded functions $ \phi $ with bounded $ \nabla \phi $, $ \Delta \phi $ to obtain an error estimate on a functional that separates measures. This comment was added as Remark 3.9 in the revised version.
> > 3. The dependence of the error bound on the dimension was clarified in the revised version, see in particular the formulation of Theorem 3.7, as well as Remarks 3.6 and 3.11.
> >
> > **Response to requested changes**.
> > The current paper measures the error between $m_0$ and $ \mu^\*\_{T-\delta} $. The empirical measure $ m_0^N $ intervenes implicitly in the construction of $ \mu^\*\_{T-\delta} $: note that $ \mu^\*\_{T-\delta} $ is the law of the time-reversed diffusion process with the learned score function $ S^\*\_{T-t} $ which is chosen to match the empirical score $ m^N_{T-t} $ (see Assumption 3.2 and Remark 3.8 of the revised version).
> >
> > We thank the Reviewer for the valuable feedback.
> > We hope that the revision satisfactorily addresses all concerns.

---

> ### Comment · Reviewer_2Vhg · 2026-03-14
> **Revision**
>
> I am satisfied with the revision, and will recommend acceptance, but I need a small clarification. Does the whole paper apply to MMD with Gaussian kernel? I am asking because it is not a linear functional. I am referring to remark 3.10

---

> > ### Comment · Reviewer_2Vhg · 2026-04-03
> > **Reminder**
> >
> > Kind reminder for my question above.

---

> > > ### Author Response · Authors · 2026-04-03
> > >
> > > Thank you for the kind reminder, and for this important clarification request.
> > > The continuous-time result, namely Theorem 3.7, does apply to MMD, including the case of the Gaussian kernel. As noted in Remark 3.10, MMD is an example of a nonlinear functional that satisfies the assumptions considered in the paper and is of important applied interest. However, in the discrete-time setting, our main result (Theorem 3.16) is stated for linear functionals $G$. Therefore, strictly speaking, it does not directly cover MMD, since MMD is not a linear functional. We believe that the discrete-time result can be extended to nonlinear functionals such as MMD. However, to keep the paper technically focused and to emphasize the study of the scheduler and the tightness of the error bound, we chose to restrict the discrete-time analysis to linear functionals. Extending the discrete-time framework to nonlinear functionals is an interesting direction for future work.

---

### Decision · Action_Editor_sX1B · 2026-04-03

**Recommendation:** Accept with minor revision

**Additional Comments:**

While reviewers are satisfied with the revised version, they propose to reorganize the large number of remarks, e.g., by moving technical remarks to the end.

**Audience:**

Yes

**Audience Explanation:**

The work is highly relevant to the generative modeling community, both for the theoretical analysis of diffusion models and the design of optimal schedulers.

**Claims And Evidence:**

Yes

**Claims Explanation:**

The theoretical error analysis seems relevant, novel, and sound. Additionally, the results are empirical validated and provide a new perspective on optimal time-scheduling.

---

> ### Author Response · Authors · 2026-04-16
>
> We thank the Editor and the reviewers for their careful evaluation, constructive feedback, and positive decision. Following their suggestion, we have reorganized the remarks in the revised manuscript in the following way:
> 1. Remarks 3.4 and 3.6 of the previous version were grouped into one Remark 3.5 put after Assumption 3.4 of the new version.
> 2. Remarks 3.8-3.12 of the previous version were grouped into one remark Remark 3.6.
> 3. Remark 3.19 of the previous version was moved to Section 4 and grouped with Remark 4.1.